

# ForgerySleuth: Empowering Multimodal Large Language Models for Image Manipulation Detection

**Zhihao Sun**[1,2]**, Haoran Jiang**[1,2]**, Haoran Chen**[1,2]**, Yixin Cao**[1,2]**, Xipeng Qiu**[1,2]**,
Zuxuan Wu**[1,2†]**, Yu-Gang Jiang**[1,2]

[1]Shanghai Key Lab of Intell. Info. Processing, School of CS, Fudan University
[2]Shanghai Collaborative Innovation Center of Intelligent Visual Computing

https://github.com/sunzhihao18/ForgerySleuth

## Abstract

Multimodal large language models have unlocked new possibilities for various multimodal tasks. However, their potential in image manipulation detection remains unexplored. When directly applied to the IMD task, M-LLMs often produce reasoning texts that suffer from hallucinations and overthinking. To address this, we propose ForgerySleuth, which leverages M-LLMs to perform comprehensive clue fusion and generate segmentation outputs indicating specific regions that are tampered with. Moreover, we construct the ForgeryAnalysis dataset through the Chain-of-Clues prompt, which includes analysis and reasoning text to upgrade the image manipulation detection task. A data engine is also introduced to build a larger-scale dataset for the pre-training phase. Our extensive experiments demonstrate the effectiveness of ForgeryAnalysis and show that ForgerySleuth significantly outperforms existing methods in generalization, robustness, and explainability.

## 1 Introduction

Recent advancements in multimodal large language models (M-LLMs) [1] have unlocked new possibilities across a variety of tasks, propelling advancements in numerous traditional research fields [2, 3, 4]. Despite these advancements, few attempts have been made to enhance the ability of multimodal large language models on the image manipulation detection task (IMD). In fact, with the rise of advanced image generation and editing tools, both creative and malicious alterations to visual content have become increasingly difficult to detect. As a result, identifying manipulated images has become essential for maintaining the integrity of digital media [5, 6, 7].

While appealing, directly applying M-LLMs for IMD produces results in text, which falls short even compared with traditional IMD methods that are able to generate segmentation masks to highlight tampered regions. In addition, M-LLMs are often prone to hallucinations and overthinking, making them unreliable for precise manipulation detection. For instance, as shown in Figure 1 (a), M-LLMs may incorrectly identify non-tampered regions, such as the "fish" area, as altered. This highlights the need to enhance the reasoning capabilities of M-LLMs for tampered region detection. Furthermore, M-LLMs, pre-trained on vast datasets, excel in recognizing authentic image patterns and world knowledge, yet lack the ability to combine low-level statistical features to pinpoint manipulation evidence, which has proven crucial for IMD [8, 9, 10, 11].

In this paper, we explore how to unleash the power of M-LLMs for IMD tasks. We introduce carefully designed modules that empower an M-LLM as an IMD expert, named ForgerySleuth. ForgerySleuth

---

†: corresponding author.

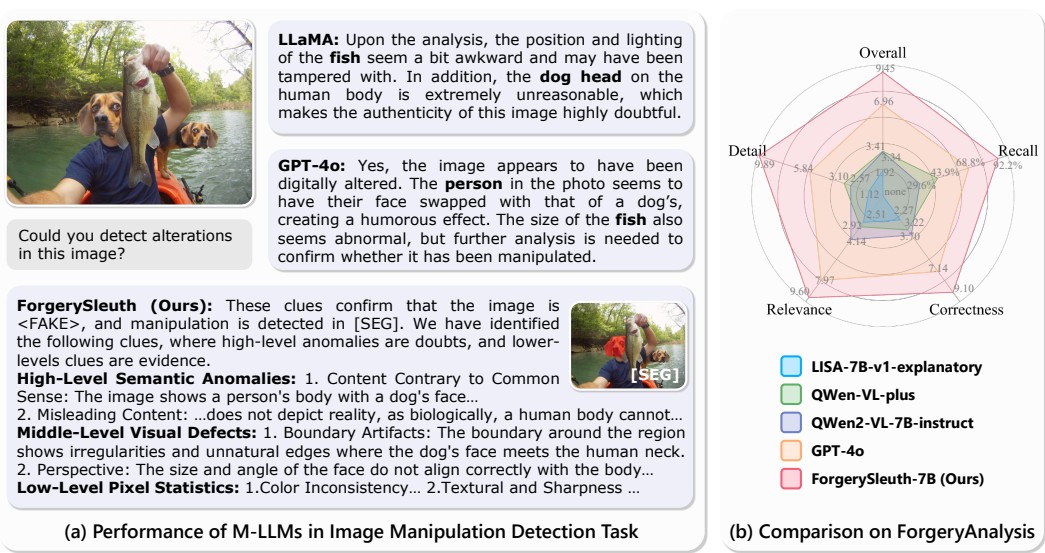

(a) Performance of M-LLMs in Image Manipulation Detection Task

(b) Comparison on ForgeryAnalysis

Figure 1: Performance and comparison of existing M-LLMs on the image manipulation detection task. Our ForgerySleuth assistant provides explanatory analysis with Chain-of-Clues and demonstrates the best forgery analysis capabilities.

is expected to provide a textual explanation of detected clues with the reasoning process, along with a segmentation mask to highlight tampered regions. Inspired by the fact that existing IMD methods rely on low-level features, such as noise patterns, to localize precise tampered regions, we aim to capture similar low-level features through ForgerySleuth. To do so, ForgerySleuth integrates M-LLMs with a trace encoder, enabling the model to leverage world knowledge to detect high-level semantic anomalies while still capturing low-level forgery traces. Additionally, inspired by LISA [12], we introduce a vision decoder with a fusion mechanism that uses attention to combine high-level anomalies in the LLM tokens and low-level traces in the trace embedding, ensuring the generation of accurate segmentation masks.

To further enhance the reasoning capability of M-LLMs on manipulation detection, we propose a supervised fine-tuning (SFT) dataset ForgeryAnalysis specifically tailored for the IMD task. Each entry in ForgeryAnalysis is initially generated by GPT-4o using a novel Chain-of-Clues prompt. Specifically, we ask GPT-4o to produce a detailed thought process and provide reasoning for why a particular region is tampered with, including high-level semantic anomalies (*e.g.*, content that contradicts common sense), mid-level visual defects (such as lighting inconsistencies), and low-level pixel information (such as color, texture, *etc.*). The generated entries are then reviewed and refined by experts. Additionally, we build a data engine based on this dataset to automate forgery analysis, enabling us to create a larger-scale ForgeryAnalysis-PT dataset for pre-training.

Extensive experiments on popular benchmarks demonstrate the success of ForgerySleuth in pixel-level manipulation localization and text-based forgery analysis. Specifically, our approach outperforms the current SoTA method by up to 24.7% in pixel-level localization tasks. Moreover, in the ForgeryAnalysis-Eval comprehensive scoring, our method surpasses the best available model, GPT-4o, achieving an improvement of 35.8%. In summary, our main contributions include:

- **Novel Exploration.** We explored the role of M-LLMs in image manipulation detection, upgrading the manipulation detection task by incorporating clues analysis and reasoning.

- **Valuable Dataset.** We constructed ForgeryAnalysis dataset, providing instructions for analysis and reasoning through Chain-of-Clues prompting. Additionally, we developed a data engine to automate forgery analysis, enabling the creation of a large-scale dataset.

- **Practical Framework.** We introduced ForgerySleuth assistant framework, which integrates M-LLMs with a trace encoder to leverage multi-level clues. The vision decoder with a fusion mechanism enables comprehensive clues fusion and segmentation outputs.

## 2 Related Work

### 2.1 LLMs and Multimodal LLMs

The success of large language models in various natural language processing tasks has led researchers to explore their integration with vision modalities, resulting in the development of M-LLMs. BLIP-2 [13] introduces a visual encoder to process image features. LLaVA [14] aligns image and text features to achieve comprehensive visual and language understanding. Researchers also utilize prompt engineering to connect independent vision and language modules via API calls without end-to-end training [15, 16, 17]. However, while these approaches enable M-LLMs to perceive, the intersection with vision-centric tasks, such as segmentation, remains underexplored. Additionally, VisionLLM [18] and LISA [12] effectively integrate segmentation capabilities into M-LLMs, making them support vision-centric tasks, such as segmentation.

With advancements in fundamental reasoning and multimodal information processing, M-LLMs have demonstrated impressive proficiency across a diverse range of tasks, including image captioning and video understanding [1]. Moreover, M-LLMs have been developed to address more complex real-world tasks in robotics, such as embodied agents [19, 20] and autonomous driving [4]. However, integrating M-LLMs into the field of image manipulation detection remains unexplored. While M-LLMs possess valuable world knowledge and can potentially detect high-level anomalies, M-LLMs are often prone to hallucinations and overthinking, making them unreliable for precise manipulation detection. Furthermore, there is no existing IMD dataset with analysis instructions for supervised fine-tuning, which further restricts their capabilities.

### 2.2 Image Manipulation Detection

Image manipulation detection is a critical task in digital image forensics. The task has evolved beyond merely determining whether an image is authentic [21, 22]; it involves localizing tampered regions and providing segmentation masks [23, 24], which leads to more intuitive results. Early attempts [25, 26, 27, 28, 29] identified anomalies and designed corresponding hand-engineered features. These efforts systematically use various tampering clues, laying a solid foundation for the field. However, such hand-engineered features are specific to certain tampering types, which limits their applicability in real-world scenarios.

Recent approaches have shifted to a more general capability of identifying complex and unknown manipulations. Semantic-agnostic features, less dependent on specific content, are thought to provide better generalization [10, 11]. Common strategies include incorporating filters or extractors to capture low-level noise features [8, 9, 30, 31] and high-frequency features [32], and using content features extracted from the image view as a supplement to detect manipulation traces [10, 30, 31]. However, many of these features are learned implicitly by the network, which limits their explainability. Studies also [11, 32] detect anomalies by comparing patch-level or object-level features. However, capturing high-level semantic anomalies, such as content that conflicts with common sense or physical laws, is still challenging. In this work, we extend the task by presenting a reasoning process with multiple levels of clues expressed in natural language, making the detection results more comprehensible. Our proposed ForgerySleuth framework leverages M-LLMs to address this challenge, effectively leveraging world knowledge to detect high-level semantic anomalies while still capturing low-level forgery traces using a trace encoder. FakeShield [33] is a concurrent work that proposes a multimodal large model for image manipulation detection.

## 3 ForgeryAnalysis Dataset

Our goal is to leverage existing M-LLMs to construct a high-quality dataset for IMD. We first describe how M-LLMs are utilized to generate initial clue analyses from various types of manipulated image sources in Section 3.1. These analyses are then meticulously refined by experts to create a high-quality dataset of 2,370 samples, which are used for the supervised fine-tuning (SFT) phase and the evaluation of M-LLMs. Section 3.2 describes our proposed data engine, which expands the 2k high-quality analysis instructions from Section 3.1 to 50k. This expansion supports the pre-training phase while maintaining quality standards. Detailed statistics of our dataset, along with examples of tampering analysis instructions, are provided in Section A.

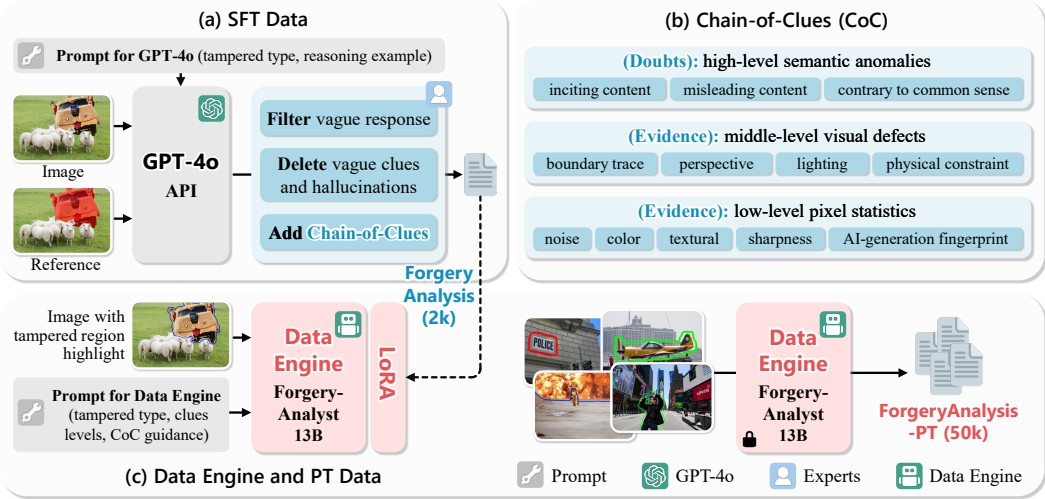

Figure 2: **ForgeryAnalysis Dataset Construction Pipeline.** Our pipeline begins with (a) GPT-4o generating initial analyses for manipulated images with annotated regions, followed by human expert review. The refined analyses are organized into (b) the Chain-of-Clues format. This human-curated ForgeryAnalysis (2k) dataset is used to train a data engine. Finally, (c) this data engine generates ForgeryAnalysis-PT, a larger-scale dataset for model pre-training.

## 3.1 ForgeryAnalysis Data

Due to the absence of instruction-following datasets that support clue analysis, we introduce the ForgeryAnalysis dataset to support the supervised fine-tuning (SFT) and evaluation of M-LLMs. As illustrated in Figure 2 (a), for each image selected from various data sources, we utilize the advanced M-LLM, GPT-4o, to generate initial clue analyses. Experts then carefully revise these analyses to eliminate hallucinations and ensure high quality. Furthermore, we have integrated a Chain-of-Clues structure to enhance reasoning capabilities.

**Data Source.** To ensure a diverse dataset that covers various types of manipulation, we collect 4,000 tampered images from existing IMD datasets, including MIML [34], CASIA2 [35], DEFACTO [36], and AutoSplice [37]. Each image is accompanied by annotations indicating the tampered regions. These sources include common real-world manipulation techniques such as splicing, copy-move, object removal, AI generation, and Photoshop edits.

**Instruction Design.** We aim to provide highly specific and content-driven descriptions of clues for each image, rather than vague or general analysis. We refer to the clues and evidence widely used in digital image forensics and manipulation detection [38, 39] to design a more reliable and effective instruction structure. The key clues include but are not limited to 1) low-level pixel statistics (*e.g.*, noise, color, texture, sharpness, and AI-generation fingerprints), 2) middle-level visual defects (*e.g.*, traces of tampered boundaries, lighting inconsistencies, and perspective relationships), and 3) high-level semantic anomalies (*e.g.*, content that contradicts common sense, incites, or misleads).

The prompt instructs GPT-4o to assume it has detected manipulation in the highlighted region of the reference image and to analyze its detection based on clues from various levels and aspects. We provide the tampering type to help GPT-4o focus on relevant clues, along with a reasoning example to guide the content and format of the output. We also instruct GPT-4o to incorporate corresponding world knowledge, such as well-known individuals or landmarks.

**Chain-of-Clues.** The responses generated by GPT-4o are subsequently revised by experts to ensure quality. Responses lacking detailed content analysis are filtered out, retaining only high-quality responses as draft data. The experts carefully review vague and incorrect statements that may arise from hallucinations, removing irrelevant clues and evidence. Inspired by works such as [40, 41] that introduce Chain-of-Thought (CoT) prompting and demonstrate its effectiveness in enhancing the step-by-step reasoning capabilities of LLMs, we propose a Chain-of-Clues (CoC) prompting approach, illustrated in Figure 2 (b). The reasoning process begins with "unveiling doubts" using

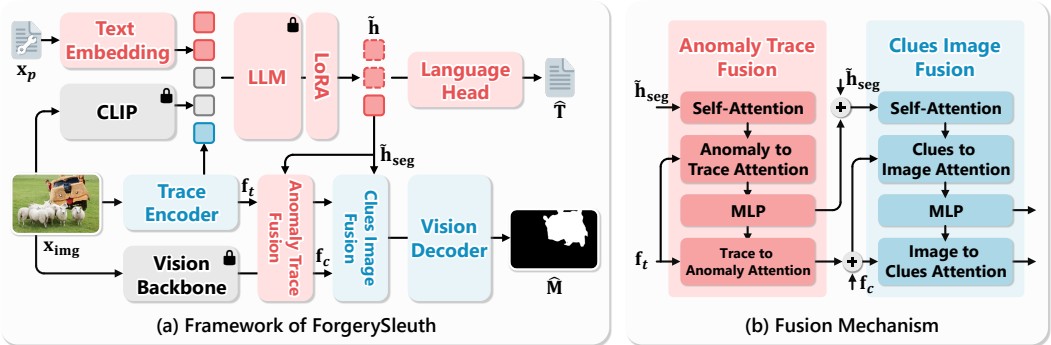

Figure 3: **Framework of ForgerySleuth.** Given an image $\mathbf{x}_{img}$ and a prompt query $\mathbf{x}_p$, the M-LLM $\mathcal{F}_m$ detects high-level semantic anomalies and generates a textual output $\hat{\mathbf{T}}$. The trace encoder $\mathcal{F}_t$ captures low-level, semantic-agnostic features. The vision decoder $\mathcal{D}$ fuses vision embeddings with the prompt embedding corresponding to [SEG] token to generate the segmentation mask $\hat{\mathbf{M}}$. LoRA is utilized in trainable modules for fine-tuning.

high-level clues, followed by "pinpointing evidence" through middle-level and low-level features. We organize the clues according to this structure, creating a coherent chain of clues, which results in 2,370 manually revised samples. To ensure accuracy, we conduct additional cross-validation with more than two experts, selecting 618 samples for evaluation. The remaining data are used for supervised fine-tuning.

## 3.2 ForgeryAnalyst Engine

The pre-training phase requires a larger-scale dataset compared to the supervised fine-tuning phase. However, the high cost of GPT-4o and the need for expert revisions make expanding the dataset a challenge. To overcome this limitation and create a pre-training dataset, we introduce an additional data engine [42, 43, 44, 45] that scales up the training set. Specifically, we fine-tune LLaVA-v1.5-13B using LoRA [46] on the ForgeryAnalysis-SFT dataset, aiming to replicate the ability of GPT-4o to analyze clues while having experts eliminate any hallucinations. The resulting model is used as our data engine that is able to annotate data automatically. In particular, we select 50k images from existing public datasets. The data engine receives input that includes explicit information about the tampered region, aiming to generate more precise and comprehensive clue analyses. It outputs manipulation analyses organized in the CoC format, as illustrated in Figure 2 (c). For more detailed information on the data engine, including the specific prompts used, please refer to Section A.2, and Figure 17 for output examples. The entire analysis generation process takes approximately 16 A800 GPU days. We refer to this dataset as ForgeryAnalysis-PT.

## 4 ForgerySleuth

Given an input image for detection and a prompt specifying the detection request, our goal is to output a binary segmentation mask $\hat{\mathbf{M}}$ of the tampered regions, as well as text $\hat{\mathbf{T}}$ that includes reasoning and evidence. This poses a challenge as the model needs to bridge vision and language modalities and capture tampering features across multiple levels. To address this, ForgerySleuth uses M-LLMs to connect vision and language modalities, enabling the description of detected clues and evidence in textual form. Additionally, we incorporate a vision decoder into the multimodal large language models to perform dense prediction, generating the tampered region mask for this specific task. The pipeline of our framework is illustrated in Figure 3 (a).

More formally, we extend the original LLM vocabulary with new tokens, including [SEG], which requests segmentation output, and <REAL> and <FAKE>, which indicate the image classification results. Given the input image $\mathbf{x}_{img}$ to be detected along with the prompt $\mathbf{x}_p$, we first feed them into the M-LLM $\mathcal{F}_m$, which outputs the hidden embedding $\tilde{h}$ at the last layer of LLM. We then extract the embedding $\tilde{\mathbf{h}}_{seg}$ corresponding to the [SEG] token from the hidden embedding $\tilde{\mathbf{h}}$ and apply an

MLP projection layer $\gamma$ to obtain $\mathbf{h}_{\text{seg}}$. The process can be formulated as

$$\tilde{\mathbf{h}} = \mathcal{F}'_m(\mathbf{x}_{\text{img}}, \mathbf{x}_p), \tag{1}$$

$$\mathbf{h}_{\text{seg}} = \gamma(\tilde{\mathbf{h}}_{\text{seg}}). \tag{2}$$

$\mathcal{F}'_m$ denotes M-LLM without the language head. This embedding represents the suspicious anomalous regions the M-LLM detects and serves as the input to guide the subsequent vision decoder.

Inspired by existing IMD methods, we propose an independent trace encoder $\mathcal{F}_t$ to focus on low-level features, which complements the relatively high-level vision and semantic features discovered by the M-LLM and provides more reliable tampering evidence, helping to minimize hallucinations from the LLM. Specifically, constrained convolutions [47] are employed with residual connections [48] at the front part of the encoder to suppress the image content and learn manipulation features adaptively. The input image $\mathbf{x}_{\text{img}}$ is also fed into encoder $\mathcal{F}_t$, producing dense manipulation trace features $\mathbf{f}_t$, which can be expressed by

$$\mathbf{f}_t = \mathcal{F}_t(\mathbf{x}_{\text{img}}). \tag{3}$$

The vision backbone $\mathcal{F}_v$ simultaneously extracts dense vision content feature $\mathbf{f}_c$ to support more precise segmentation. Finally, $\mathbf{f}_c$, $\mathbf{f}_t$ and $\mathbf{h}_{\text{seg}}$ are fed to the vision decoder $\mathcal{D}$ to generate the segmentation mask $\hat{\mathbf{M}}$ which indicates the tampered region. The language modeling head $\mathcal{H}$ processes $\tilde{\mathbf{h}}$ and outputs $\hat{\mathbf{y}}_{\text{txt}}$, containing an analysis of the reasoning and evidence. It can be formulated as

$$\mathbf{f}_c = \mathcal{F}_v(\mathbf{x}_{\text{img}}), \tag{4}$$

$$\hat{\mathbf{M}} = \mathcal{D}(\mathbf{f}_c, \mathbf{f}_t, \mathbf{h}_{\text{seg}}), \quad \hat{\mathbf{T}} = \mathcal{H}(\tilde{\mathbf{h}}). \tag{5}$$

**Fusion Mechanism.** To integrate the image content embedding $\mathbf{f}_c$, trace embedding $\mathbf{f}_t$, and the LLM output tokens $\mathbf{h}_{\text{seg}}$ obtained from the M-LLM, we take inspiration from Transformer segmentation models [49, 50, 42] and design a vision decoder with a fusion attention mechanism, as illustrated in Figure 3 (b). Here, the first layer of the module computes attention between anomalies in the LLM output tokens and traces in the trace embeddings, facilitating the organization and pinpointing of clues. The subsequent layers focus on attention between refined clues in the upgraded tokens and content in the image embeddings, enabling more precise segmentation of the tampered regions.

**Learning Objective.** Our framework is trained end-to-end using both the reasoning text loss $\mathcal{L}_{\text{txt}}$ and the tampered region mask loss $\mathcal{L}_{\text{mask}}$. The final learning objective $\mathcal{L}$ is formulated as a weighted sum of these losses, with weight $\lambda_{\text{txt}}$ and $\lambda_{\text{mask}}$, as

$$\mathcal{L} = \lambda_{\text{txt}}\mathcal{L}_{\text{txt}} + \lambda_{\text{mask}}\mathcal{L}_{\text{mask}}. \tag{6}$$

Specifically, $\mathcal{L}_{\text{txt}}$ is the auto-regressive cross-entropy (CE) loss for reasoning text generation, guiding the model to collect multi-level clues, while $\mathcal{L}_{\text{mask}}$ is the mask loss, which encourages the model to produce precise segmentation results. $\mathcal{L}_{\text{mask}}$ is implemented by the combination of binary cross-entropy (BCE) loss and DICE loss, with respective weights $\lambda_{bce}$ and $\lambda_{\text{dice}}$. Given the ground-truth targets $\mathbf{T}$ and $\mathbf{M}$, the final loss functions are defined as

$$\mathcal{L}_{\text{txt}} = \mathbf{CE}(\hat{\mathbf{T}}, \mathbf{T}), \tag{7}$$

$$\mathcal{L}_{\text{mask}} = \lambda_{\text{bce}}\mathbf{BCE}(\hat{\mathbf{M}}, \mathbf{M}) + \lambda_{\text{dice}}\mathbf{DICE}(\hat{\mathbf{M}}, \mathbf{M}). \tag{8}$$

It is noteworthy that no separate loss is required for the classification result, as the supervision for <REAL> and <FAKE> tokens are integrated into the text loss $\mathcal{L}_{\text{txt}}$.

## 4.1 Training Strategy

**Trainable Parameter.** To preserve the world knowledge and the normal patterns of authentic images learned by the pre-trained M-LLM $\mathcal{F}_m$, we adopt LoRA [46] for efficient fine-tuning. The vision backbone $\mathcal{F}_v$ is entirely frozen to retain the capacity for modeling image content features, which are crucial for accurate segmentation. Both the trace encoder $\mathcal{F}_t$ and the vision decoder $\mathcal{D}$ are fully trainable and fine-tuned. Additionally, the token embeddings of the LLM, the language modeling head $\mathcal{H}$, and the projection layer $\gamma$ are updated during training. Despite the large model scale, only 5.47% of the parameters are trainable, making end-to-end training more efficient.

Table 1: Manipulation localization results comparing ForgerySleuth with SoTA methods.

| Method | Optimal Threshold F1 | | | | | Fixed Threshold F1 (0.5) | | | | |
| --- | --- | --- | --- | --- | --- | --- | --- | --- | --- | --- |
| | Columbia | Coverage | CASIA1 | NIST16 | COCOGlide | Columbia | Coverage | CASIA1 | NIST16 | COCOGlide |
| Mantra-Net [8] | 0.650 | 0.486 | 0.320 | 0.225 | 0.673 | 0.508 | 0.317 | 0.180 | 0.172 | 0.516 |
| SPAN [9] | 0.873 | 0.428 | 0.169 | 0.363 | 0.350 | 0.759 | 0.235 | 0.112 | 0.228 | 0.298 |
| MVSS-Net [10] | 0.781 | 0.659 | 0.650 | 0.372 | 0.642 | 0.729 | 0.514 | 0.528 | 0.320 | 0.486 |
| PSCC-Net [53] | 0.760 | 0.615 | 0.670 | 0.210 | 0.685 | 0.604 | 0.473 | 0.520 | 0.113 | 0.515 |
| CAT-Net2 [54] | 0.923 | 0.582 | 0.852 | 0.417 | 0.603 | 0.859 | 0.381 | 0.752 | 0.308 | 0.434 |
| TruFor [30] | 0.914 | 0.735 | 0.822 | 0.470 | 0.720 | 0.859 | 0.600 | 0.737 | 0.399 | 0.523 |
| UnionFor. [31] | 0.925 | 0.720 | 0.863 | 0.489 | 0.742 | 0.861 | 0.592 | 0.760 | 0.413 | 0.536 |
| **ForgerySleuth** | **0.931** | **0.792** | **0.870** | **0.610** | **0.751** | **0.925** | **0.684** | **0.804** | **0.518** | **0.562** |

**Data Formulation.** The training process involves two phases: **1) Pre-training Phase**: (a) In the first stage, we aim to align the framework modules and ensure that the model can perform basic segmentation and visual reasoning tasks. For foundational segmentation abilities, we use semantic segmentation datasets such as ADE20k [51] and COCO-Stuff [52], transforming these datasets into visual question-answer pairs using class names as questions. Additionally, we incorporate ReasonSeg [12] to strengthen visual reasoning capabilities. (b) The ForgeryAnalysis-PT dataset focuses on the IMD task, including analysis instructions that enable the model to recognize tampering traces and identify clues. We also utilize public IMD datasets, including MIML [34] and CASIA2 [35], with prompts randomly selected from simple or vague responses. **2) Supervised Fine-tuning Phase**: The ForgeryAnalysis-SFT dataset, meticulously revised by experts to ensure the accuracy of reasoning and analysis, is used for final supervised fine-tuning to standardize the analysis and output.

# 5 Experiment

## 5.1 Experimental Setting

**Testing Dataset.** We utilize six publicly accessible test datasets, which are Columbia [55], Coverage [56], CASIA1 [35], NIST16 [57], IMD20 [58], and COCOGlide [30], to evaluate and compare our method with state-of-the-art methods thoroughly. To effectively evaluate the model's generalization capability, these datasets are ensured to be disjoint from the training data. Additionally, we use our ForgeryAnalysis-Eval dataset to assess the reasoning and analysis capabilities of the methods.

**Evaluation Metrics.** Localizing the tampered regions at the pixel level is crucial in image manipulation detection. We follow established practices [31] by using optimal threshold and fixed threshold F1 scores and the threshold-independent Area Under the Curve (AUC) metric. To ensure fairness and precision in our comparative analysis, we refer to some results for other methods from [31, 32].

Evaluating the novel reasoning analysis outputs presents a unique challenge, as it involves assessing the comprehension, reasoning, and correctness in generating text explanations. To measure the similarity between the generated text and the ground-truth text, thereby reflecting its accuracy, we incorporate Semantic Textual Similarity (STS) metric. We follow SBERT [59] and use STS to measure the similarity in our ForgeryAnalysis-Eval, where various models are utilized to calculate the similarity. Additionally, inspired by previous work [60, 14], we use GPT-4 as an automated evaluator to assess the reasoning performance of different models, which enables a more holistic evaluation of the generated analysis text, Figure 18 and Section D.3 provide the prompt format and the evaluation criteria. We conduct evaluations twice and report the average performance.

## 5.2 Manipulation Detection Results

The results in Table 1 and Table 2 demonstrate the performance of our ForgerySleuth and comparisons with SoTA methods for image manipulation localization, using pixel-level F1 scores and AUC metrics, respectively. Our method consistently achieves the highest or second-highest AUC. Regarding F1 scores, our approach surpasses other methods across all datasets, showcasing its reliability under both fixed and optimal thresholds. On challenging datasets like NIST16 and IMD20, ForgerySleuth outperforms UnionFormer by margins of 0.105 and 0.121 for fixed and optimal thresholds, respectively, which we believe is significant given its challenging nature. Furthermore, on the COCOGlide dataset, which features novel diffusion-based manipulations, our model also exceeds the UnionFormer. These significant improvements can be attributed to the capability to effectively capture both low-level

Table 2: Manipulation localization results of ForgerySleuth and SoTA methods, using pixel-level AUC as the evaluation metric.

| Method | Columbia | Coverage | CASIA1 | NIST16 | IMD20 |
|---|---|---|---|---|---|
| Mantra-Net [8] | 0.824 | 0.819 | 0.817 | 0.795 | 0.748 |
| SPAN [9] | 0.936 | 0.922 | 0.797 | 0.840 | 0.750 |
| PSCC-Net [53] | 0.982 | 0.847 | 0.829 | 0.855 | 0.806 |
| ObjectFormer [32] | 0.955 | 0.928 | 0.843 | 0.872 | 0.821 |
| TruFor [30] | 0.947 | 0.925 | 0.957 | 0.877 | - |
| UnionFormer [31] | 0.989 | 0.945 | **0.972** | 0.881 | 0.860 |
| **ForgerySleuth** | **0.992** | **0.962** | 0.969 | **0.898** | **0.911** |

Table 3: Robust evaluation results of ForgerySleuth and existing methods using pixel-level AUC.

| Distortion | SPAN | PSCC-Net | ObjectFor. | UnionFor. | **Ours** |
|---|---|---|---|---|---|
| *w/o* distortion | 0.8395 | 0.8547 | 0.8718 | 0.8813 | **0.8982** |
| Resize ($0.78\times$) | 0.8324 | 0.8529 | 0.8717 | 0.8726 | **0.8962** |
| Resize ($0.25\times$) | 0.8032 | 0.8501 | 0.8633 | 0.8719 | **0.8792** |
| GSBr ($k=3$) | 0.8310 | 0.8538 | 0.8597 | 0.8651 | **0.8863** |
| GSBr ($k=15$) | 0.7915 | 0.7993 | 0.8026 | 0.8430 | **0.8658** |
| GSN ($\sigma=3$) | 0.7517 | 0.7842 | 0.7958 | 0.8285 | **0.8452** |
| GSN ($\sigma=15$) | 0.6728 | 0.7665 | 0.7815 | 0.8057 | **0.8139** |
| JPEG ($q=100$) | 0.8359 | 0.8540 | 0.8637 | 0.8802 | **0.8974** |
| JPEG ($q=50$) | 0.8068 | 0.8537 | 0.8624 | 0.8797 | **0.8839** |

trace features and high-level semantic inconsistencies, enabling it to detect even subtle generative manipulations. Overall, the results emphasize the generalization ability of ForgerySleuth.

**Robustness Evaluation.** We apply several image distortions to the NIST16 dataset following [31, 32] to evaluate the robustness of our method and compare the results with other methods. The distortions included 1) resizing images to different scales, 2) applying Gaussian blur with different kernel sizes $k$, 3) adding Gaussian noise with various standard deviation $\sigma$, and 4) compressing images using JPEG with different quality factors $q$. The results in Table 3 show that our model consistently outperforms other methods across all types of distortions. This improvement in robustness stems from the ability to identify high-level semantic anomalies rather than relying solely on low-level statistical features that are more susceptible to distortions.

## 5.3 Forgery Analysis Results

We compare our forgery analysis results with several M-LLMs, including GPT-4o, Qwen2-VL [61], and LISA [12]. Additionally, we perform LoRA fine-tuning on the LISA using the ForgeryAnalysis-SFT to provide a more comprehensive comparison. We incorporate STS to measure the similarity between the generated text and the ground-truth text in ForgeryAnalysis-Eval. The results, shown in Table 4, confirm that our ForgerySleuth consistently outperforms other methods.

We also leverage GPT-4 as an evaluator based on the ForgeryAnalysis-Eval dataset to assess the quality of text analysis and reasoning. We collect answers from each M-LLM, and GPT-4 assigns a score from 1 to 10 for each response. Beyond scoring, GPT-4 provides detailed explanations for its ratings. We also report the recall rate to directly reflect the ability to identify tampered images.

Figure 1 (b) presents the scores of different models without additional fine-tuning across various evaluation dimensions. The existing models exhibit low recall rates, and the overall evaluation suggests that they struggle to identify manipulations and provide accurate analyses. Figure 5 shows the scores of LISA and ForgerySleuth after SFT, along with the versions without SFT. ForgerySleuth shows an improvement of 5.05 in the overall score compared to LISA, further demonstrating the effectiveness of our design targeted specifically for the IMD task. Furthermore, using the SFT dataset results in performance gains for both methods, indicating the quality of the ForgeryAnalysis dataset.

## 5.4 Ablation Study

We conduct an extensive ablation study to analyze the effect of our ForgeryAnalysis dataset and each component and setting within ForgerySleuth framework. We report the pixel-level localization

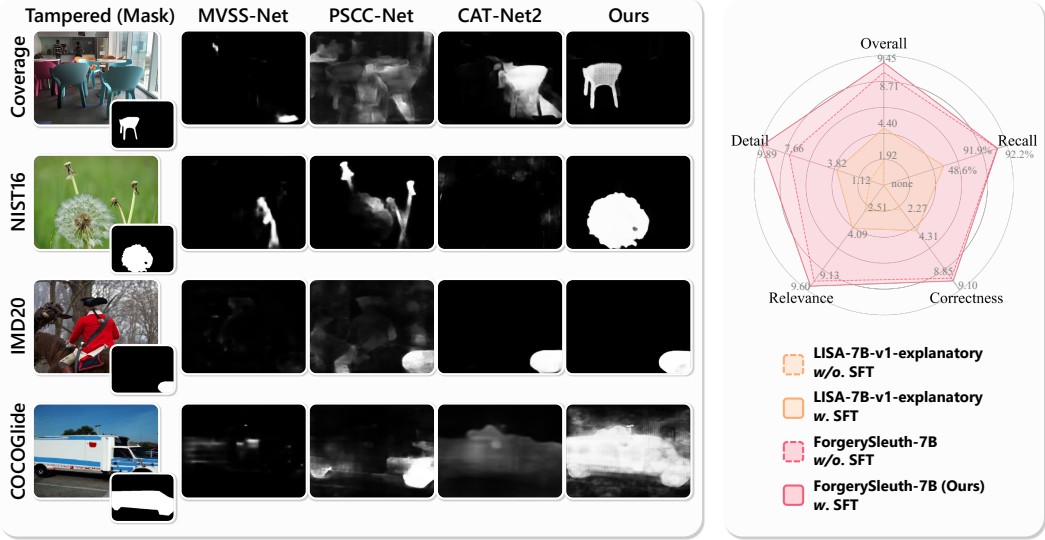

Figure 4: Visualization results comparing ForgerySleuth with existing methods. The examples are taken from various datasets.

Figure 5: Forgery analysis results of the ablation study.

Table 4: Forgery analysis results of ForgerySleuth and SoTA methods, evaluated using STS.

| Model | Dim | Ours | GPT-4o | QWen2 | QWen | LISA |
|---|---|---|---|---|---|---|
| MiniLM-L6-v2 | 384 | **0.926** | 0.725 | 0.595 | 0.551 | 0.313 |
| MiniLM-L12-v2 | 384 | **0.919** | 0.645 | 0.505 | 0.475 | 0.350 |
| mpnet-base-v2 | 768 | **0.961** | 0.724 | 0.635 | 0.546 | 0.401 |

Table 5: Ablation study on different parts of ForgeryAnalysis.

| Data | NIST16 | IMD20 | COCOGlide | ForgeryA. |
|---|---|---|---|---|
| ForgeryA.-SFT | 0.191 | 0.094 | 0.152 | 5.09 |
| ForgeryA.-PT | 0.516 | 0.705 | **0.571** | 8.71 |
| ForgeryA.-PT&SFT | **0.518** | **0.710** | 0.562 | **9.45** |

performance on IMD datasets using the F1 score. We use ForgeryAnalysis-Eval to evaluate the quality of the reasoning text, following the scoring criteria described in Section 5.3.

**Contribution of ForgeryAnalysis.** The experimental results in Table 5 show the performance drop when ForgeryAnalysis-PT data is excluded during the pre-training phase, demonstrating the importance of large-scale data. Using ForgeryAnalysis-PT, the model achieves strong performance on the IMD task, even without the final stage of supervised fine-tuning with ForgeryAnalysis-SFT data. However, this final fine-tuning further enhances the quality of the tampering analysis text.

**Effect of Designed Components.** As shown in Table 6, removing the trace encoder $\mathcal{F}_t$ significantly degrades performance across all datasets, confirming its critical contribution in capturing low-level trace features and mitigating hallucinations in the M-LLM. At the same time, the performance drop observed when the fusion mechanism is excluded highlights the effectiveness of our fusion strategy in integrating multimodal and trace-based information.

**Impact of Training Strategies.** In our comparison of various training strategies for the visual encoder $\mathcal{F}_v$, including trainable and LoRA fine-tuned, we discover that the best performance is achieved by keeping the encoder completely frozen. This could be attributed to the fact that trainable and LoRA fine-tuned strategies slightly diminish the generalization ability of the original SAM.

## 5.5 Qualitative Results

Figure 4 showcases pixel-level localization results across different datasets, comparing our framework with other SoTA methods. The masks are displayed without binarization to provide a more detailed view of the localization capability. Our method consistently delivers more precise tampered region detection with higher confidence across various types of manipulation. More detailed examples of clues and analysis provided by ForgerySleuth can be found in Section A.4. Across different types of manipulation, relevant clues and high-quality analysis demonstrate the effectiveness of M-LLM in capturing high-level semantic anomalies.

Table 6: Ablation study on different modules and settings, using pixel-level F1 with fixed threshold 0.5 as the evaluation metric.

| Setting | CASIA1 | NIST16 | IMD20 | COCOGlide |
|---|---|---|---|---|
| 1. *w/o.* Trace Enc. $\mathcal{F}_t$ | 0.637 | 0.323 | 0.622 | 0.395 |
| 2. *w/o.* Fusion Mechanism | 0.628 | 0.463 | 0.649 | 0.513 |
| 3. *w.* $\mathcal{F}_v$ Trainable | 0.755 | 0.451 | 0.716 | 0.526 |
| 4. *w.* $\mathcal{F}_v$ LoRA-ft. | 0.766 | 0.493 | **0.731** | 0.547 |
| **ForgerySleuth** | **0.804** | **0.518** | 0.710 | **0.562** |

## 6 Conclusion

In this work, we explored the potential of multimodal large language models in the image manipulation detection task. The proposed ForgerySleuth integrates M-LLMs with a trace encoder, allowing the model to utilize world knowledge to detect high-level semantic anomalies while effectively capturing low-level forgery traces. Additionally, we introduced a vision decoder with a fusion mechanism to integrate different features, ultimately producing precise segmentation masks. We also proposed a supervised fine-tuning dataset, ForgeryAnalysis, specifically designed for the IMD task. Each entry was initially generated by GPT-4o using a novel Chain-of-Clues prompt and then reviewed and refined by experts. Furthermore, we developed a data engine based on this dataset to automate forgery analysis, facilitating the creation of a larger-scale ForgeryAnalysis-PT dataset for pre-training purposes. A discussion of limitations can be found in Section E. We have already made the resources publicly available, including the data, code, and weights, to provide resources for advancing the field.

**Acknowledgment.** This work was supported in part by the National Natural Science Foundation of China (Grant 62472098). We would like to express our gratitude to Ruirui Tu and Xu Han for their valuable assistance with data annotation, which significantly contributed to the development of the ForgeryAnalysis dataset.

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

# A  ForgeryAnalysis Dataset

## A.1  ForgeryAnalysis Data

We utilize the advanced M-LLM, GPT-4o, to generate the initial clue analyses, carefully designing prompts to ensure GPT-4o provides accurate and detailed responses. First, we inform GPT-4o of its role as an assistant, clearly outlining the levels of clues, along with specific examples, and specifying the task it needs to complete. The detailed prompt is as follows:

[ROLE] You are a rigorous and responsible image tampering (altering) detection expert. You can detect whether an image has been tampered with, localize the exact tampered region, and analyze your detection decision according to tampering clues of different levels. These clues include but are not limited to low-level pixel statistics (such as noise, color, textural, sharpness, and AI-generation fingerprint), middle-level visual defects (such as traces of tampered region or boundary, lighting inconsistency, perspective relationships, and physical constraints), and high-level semantic anomalies (such as content contrary to common sense, inciting and misleading content), etc. Altering operations could be divided into types, including "splice", "copy-move", "remove", and "AI-generate", leaving different clues that you should consider.

[TASK] Now, your task is to provide analysis. Please note that in real detection scenarios, you cannot know in advance whether an image has been tampered with and the specific tampered region. However, now I will tell you this information to help you conduct a more rigorous and accurate analysis based on this. There is no need to include all aspects and views in your analysis, give some of your most confident points.

The details of each conversation round are illustrated in Figure 12. In each round, we provide two images: the tampered image to be analyzed and a reference image with the tampered region highlighted. The prompt includes the <FAKE> token to indicate that the image is manipulated, specific tampering types to help the model focus on relevant clues, and a structured clue analysis format. Based on the response of GPT-4o, experts then conduct further revisions. The experts carefully review vague and incorrect statements that may arise from hallucinations, removing irrelevant clues and evidence. They also reorganize the clues into the Chain-of-Clues structure, which guides the reasoning process. This begins with "unveiling doubts" using high-level clues and continues with "pinpointing evidence" using middle-level and low-level features. The experts also check special tokens, such as <FAKE> and [SEG], to meet the requirements for subsequent model training.

## A.2  ForgeryAnalyst Engine

It is worth noting that the M-LLMs used in the data engine ForgeryAnalyst and the detection framework ForgerySleuth are different and independent. Although both M-LLMs are designed to analyze forgery and produce text-based clue analyses, their inputs and tasks are distinct. ForgeryAnalyst receives input that includes explicit information about the tampered region (highlighted to indicate tampering) with the goal of generating more precise and comprehensive clue analyses to construct pre-training data. In contrast, ForgerySleuth takes an image to be analyzed, aiming for the M-LLM to identify high-level semantic anomalies for detecting tampered regions. Furthermore, we design different prompts tailored to these two specific tasks. In our experiments, ForgeryAnalyst employs LLaVA-v1.5-13B, while the MLLM in ForgerySleuth uses LLaVA-7B-v1-1, balancing performance and efficiency.

We also designed a dedicated prompt for the data engine. In addition to indicating the image type as <FAKE> and specifying the tampering type [MANIPULATION-TYPE], we provide detailed instructions and examples of the Chain-of-Clues (CoC), as well as the required output data format. The specific prompt format is as follows:

You are a rigorous and responsible image tampering (altering) detection expert. You can localize the exact tampered region and analyze your detection decision according to tampering clues at different levels. Assuming that you have detected this is a <FAKE> image and the manipulation type is [MANIPULATION-TYPE], the exact tampered region boundary is highlighted with color in this image (and your detection IS correct).

Table 7: Statistic of our ForgeryAnalysis Dataset.

| Dataset | Split | Count | Sum |
|---|---|---|---|
| ForgeryAnalysis | ForgeryAnalysis-Eval | 618 | 2,370 |
| | ForgeryAnalysis-SFT | 1752 | |
| ForgeryAnalysis-PT | - | 50k | 50k |

Table 8: Overview of public dataset utilized in the construction of ForgeryAnalysis and for evaluation.

| Dataset | Usage | Authentic | Tampered | Manipulation Types | Source |
|---|---|---|---|---|---|
| MIML [34] | Train, Construct | 11,142 | 123,150 | Manual Editing | PSBattles |
| CASIA2 [35] | Train, Construct | 7,491 | 5,063 | Splice, Copy-move | Corel |
| DEFACTO [36] | Train, Construct | - | 149,587 | Splice, Copy-move, AIGC (GAN) | MS COCO |
| AutoSplice [37] | Train, Construct | 2,273 | 3,621 | AIGC | Visual News |
| Columbia [55] | Eval | 183 | 180 | Splice | - |
| Coverage [56] | Eval | 100 | 100 | Copy-move | - |
| CASIA1 [35] | Eval | 800 | 920 | Splice, Copy-move | Corel |
| NIST16 [57] | Eval | - | 564 | Splice, Copy-move | - |
| IMD20 [58] | Eval | 414 | 2,010 | Manual Editing | - |
| COCOGlide [30] | Eval | 512 | 512 | AIGC (Diffusion) | - |

Please provide the chain-of-clues supporting your detection decision in the following style: # high-level semantic anomalies (such as content contrary to common sense, inciting and misleading content), # middle-level visual defects (such as traces of tampered region or boundary, lighting inconsistency, perspective relationships, and physical constraints) and # low-level pixel statistics (such as noise, color, textural, sharpness, and AI-generation fingerprint), where the high-level anomalies are significant doubts worth attention, and the middle-level and low-level findings are reliable evidence.

## A.3 Statistics

Table 7 presents the data statistics of the ForgeryAnalysis dataset. ForgeryAnalysis-Eval and ForgeryAnalysis-SFT are initially generated by GPT-4o and fully revised by experts. They are used for evaluating the quality of manipulation analysis generated by the M-LLMs and for the final supervised fine-tuning, respectively. ForgeryAnalysis-PT is automatically constructed by our proposed data engine, ForgeryAnalyst, maintaining consistency in data format with the other subsets. Table 8 provides detailed statistics on the scale, source, and manipulation types from each public dataset utilized in the construction of ForgeryAnalysis and for evaluation.

## A.4 Cases

Figures 13, 14, 15 and 16 present examples of analysis texts from ForgeryAnalysis-Eval for four different tampering types. These diverse cases highlight the variations in detectable clues across different tampering types and illustrate how varying levels of clues support manipulation detection. The analyses in ForgeryAnalysis-Eval are cross-checked by multiple experts to ensure comprehensive and accurate clues. Figure 17 shows analysis texts in ForgeryAnalysis-PT. Although the subset is automatically generated by the data engine, it also provides precise descriptions and analyses of the tampered regions.

# B  ForgerySleuth

## B.1  Framework Details

**Trace Encoder.** Considering that the vision backbone (ViT-H SAM backbone) is pre-trained on tasks highly correlated with semantics and remains frozen during the fine-tuning of our framework, the semantic-agnostic features widely used in IMD tasks [8, 9, 10, 11] cannot be effectively leveraged. We propose an independent trace encoder $\mathcal{F}_t$, equipped with a noise enhancement module to focus on low-level features and provide more reliable tampering evidence. Specifically, the noise enhancement module, positioned at the front of the encoder, uses constrained convolutions [47] to compute local

Table 9: Manipulation localization results comparing ForgerySleuth with SoTA methods.

| Method | Optimal Threshold F1 | | | | | Fixed Threshold F1 (0.5) | | | | |
|---|---|---|---|---|---|---|---|---|---|---|
| | Columbia | Coverage | CASIA1 | NIST16 | COCOGlide | Columbia | Coverage | CASIA1 | NIST16 | COCOGlide |
| Mantra-Net [8] | 0.650 | 0.486 | 0.320 | 0.225 | 0.673 | 0.508 | 0.317 | 0.180 | 0.172 | 0.516 |
| SPAN [9] | 0.873 | 0.428 | 0.169 | 0.363 | 0.350 | 0.759 | 0.235 | 0.112 | 0.228 | 0.298 |
| MVSS-Net [10] | 0.781 | 0.659 | 0.650 | 0.372 | 0.642 | 0.729 | 0.514 | 0.528 | 0.320 | 0.486 |
| PSCC-Net [53] | 0.760 | 0.615 | 0.670 | 0.210 | 0.685 | 0.604 | 0.473 | 0.520 | 0.113 | 0.515 |
| CAT-Net2 [54] | 0.923 | 0.582 | 0.852 | 0.417 | 0.603 | 0.859 | 0.381 | 0.752 | 0.308 | 0.434 |
| TruFor [30] | 0.914 | 0.735 | 0.822 | 0.470 | 0.720 | 0.859 | 0.600 | 0.737 | 0.399 | 0.523 |
| UnionFor. [31] | 0.925 | 0.720 | 0.863 | 0.489 | 0.742 | 0.861 | 0.592 | 0.760 | 0.413 | 0.536 |
| FakeShield [33] | 0.306 | 0.085 | 0.620 | 0.119 | 0.659 | 0.285 | 0.052 | 0.566 | 0.099 | 0.536 |
| - only *TP samples* | 0.874 | 0.470 | 0.696 | 0.514 | 0.659 | 0.813 | 0.289 | 0.635 | 0.431 | 0.536 |
| **ForgerySleuth** | **0.931** | **0.792** | **0.870** | **0.610** | **0.751** | **0.925** | **0.684** | **0.804** | **0.518** | **0.562** |

Table 10: Manipulation localization results of ForgerySleuth and SoTA methods, using pixel-level AUC as the evaluation metric.

| Method | Columbia | Coverage | CASIA1 | NIST16 | IMD20 |
|---|---|---|---|---|---|
| Mantra-Net [8] | 0.824 | 0.819 | 0.817 | 0.795 | 0.748 |
| SPAN [9] | 0.936 | 0.922 | 0.797 | 0.840 | 0.750 |
| PSCC-Net [53] | 0.982 | 0.847 | 0.829 | 0.855 | 0.806 |
| ObjectFor. [32] | 0.955 | 0.928 | 0.843 | 0.872 | 0.821 |
| TruFor [30] | 0.947 | 0.925 | 0.957 | 0.877 | - |
| UnionFor. [31] | 0.989 | 0.945 | **0.972** | 0.881 | 0.860 |
| FakeShield [33] | 0.323 | 0.137 | 0.787 | 0.185 | 0.780 |
| - only *TP samples* | 0.924 | 0.761 | 0.883 | 0.801 | 0.868 |
| **ForgerySleuth** | **0.992** | **0.962** | 0.969 | **0.898** | **0.911** |

differences, extract noise features, and suppress image content. The convolution kernel constraints are defined as follows:

$$\begin{cases} \omega_{(0,0)} = 1, \\ \sum_{(m,n)} \omega_{(m,n)} = 0, \end{cases} \tag{9}$$

where $(m, n)$ denotes the spatial index of the values within the convolution kernel, with $(0, 0)$ positioned at the center. The constrained convolutions are still trainable, allowing them to learn manipulation features more adaptively than fully fixed-parameter noise extractors. The extracted noise features further enhance the original features by residual connections. The encoder employs a ViT-B architecture and all parameters are fine-tuned during training.

In summary, we employ the trace encoder $\mathcal{F}_t$ to capture low-level manipulation features, leveraging constrained convolutions within the noise enhancement module to achieve this. Meanwhile, the vision backbone $\mathcal{F}_v$ utilizes pre-trained parameters from SAM, which is widely recognized for its ability to capture dense visual content features, thereby supporting more precise segmentation.

**Fusion Mechanism.** To integrate the image content embedding $\mathbf{f}_c$, trace embedding $\mathbf{f}_t$, and the LLM output tokens $\mathbf{h}_{seg}$ obtained from the M-LLM, we take inspiration from Transformer segmentation models [49, 50, 42] and design a vision decoder with a fusion attention mechanism, illustrated in Figure 3 (b). The mechanism consists of three layers, with each layer performing four steps: self-attention on the LLM output tokens or upgraded tokens, cross-attention from tokens (as queries) to the trace or image embeddings, point-wise MLP, and cross-attention from the trace or image embeddings (as queries) back to the tokens. The first layer of the module computes attention between anomalies in the LLM output tokens and traces in the trace embeddings, facilitating the organization and pinpointing of clues. The subsequent layers focus on attention between refined clues in the upgraded tokens and content in the image embeddings, enabling more precise segmentation of the tampered regions.

# C   Comparison with FakeShield

## C.1   Details

FakeShield [33] is a concurrent work with the same motivation as ours. It proposes a multimodal large model for image manipulation detection. This further reinforces the significance of providing a reasonable forgery analysis, alleviating the explainability issues in existing image manipulation detection methods. However, there are some fundamental differences between FakeShield and our approach.

From a data perspective, FakeShield relies on GPT-4o to generate tampering analysis texts, constructing "image-mask-description" triplets for training and evaluation. However, this process presents certain challenges: First, the generated texts lack structured reasoning, as FakeShield does not impose explicit guidance on how the detected clues is organized and analysis data is formulated. Second, reliance on GPT-4o alone makes data prone to hallucinations, potentially introducing unreliable or misleading explanations.

In contrast, we propose a three-level Chain-of-Clues (CoC) structure, which enables a more structured and interpretable analysis. Our design is based on two key insights: 1) Previous research [38, 39] suggests that forensic clues are naturally organized hierarchically. 2) Studies on Chain-of-Thought (CoT) prompting [40, 41] have demonstrated its effectiveness in improving reasoning within large language models. Inspired by these findings, we structure the tampering analysis data into a progressive reasoning process, encouraging the model to identify, refine, and interpret forensic clues at different levels. Additionally, to reduce hallucinations and improve reliability, we manually verify the generated analysis texts and expand the dataset using an additional data engine.

From a model perspective, FakeShield employs a multimodal forgery localization module (MFLM) that leverages the segment anything model (SAM) and a large language model to localize tampered regions based on their general vision and language features. However, recent studies [10, 54, 11] highlight the crucial role of low-level noise features in image manipulation detection, while pre-trained models for general vision tasks primarily capture high-level semantic features. To address this challenge, we introduce a trace encoder with noise enhancement to compensate for the limitations of the base vision model. In addition, we design a dedicated fusion mechanism to better integrate multiple features. Ablation studies in Table 6 further validate the effectiveness of these components.

In the following experiments, we comprehensively compare FakeShield and our method, ForgerySleuth, demonstrating that ForgerySleuth consistently achieves superior performance in various evaluation aspects.

## C.2   Evaluation

**Experimental Setting.** We reproduce the FakeShield model using its publicly released official code repository and pre-trained weights. The following comparative experiments adopt the same experimental settings and evaluation metrics as described in Section 5.1.

We follow established practices [31] by using optimal threshold and fixed threshold F1 scores and the threshold-independent Area Under the Curve (AUC) metric. Inspired by prior work [60, 14], we use GPT-4 as an automated evaluator to assess the reasoning performance of different models. To address concerns regarding potential limitations of GPT-4, we incorporate an additional metric, semantic textual similarity (STS). Specifically, following SBERT [59], we use STS to measure the similarity between the generated analysis and the ground-truth text in ForgeryAnalysis-Eval.

**Manipulation Detection Results.** Tables 9 and 10 present the performance of ForgerySleuth, FakeShield, and other SoTA methods in the manipulation localization task, reporting F1 scores and AUC metrics, respectively. It is essential to note that the official implementation of FakeShield follows a two-stage pipeline. First, it determines whether an image has been manipulated. Only if an image is classified as fake does it proceed to generate a tampering mask. In contrast, ForgerySleuth and other detection methods output a mask for all images while also providing a real/fake classification result.

We adopt the following strategy for FakeShield to maintain consistency with other methods and ensure a fair comparison. If FakeShield classifies an image as real, we assume that the predicted mask is an all-zero matrix (*i.e.*, indicating no tampered region). The corresponding results are

Table 11: Forgery analysis results of ForgerySleuth and SoTA methods, evaluated using GPT-4 on ForgeryAnalysis-Eval dataset.

| Method | *Recall* | Correctness | Relevance | Detail | Overall |
|---|---|---|---|---|---|
| LISA [12] | - | 2.27 | 2.51 | 1.12 | 1.92 |
| QWen | 43.9% | 3.22 | 2.92 | 3.10 | 3.41 |
| QWen2 [61] | 29.6% | 3.70 | 4.14 | 2.57 | 3.34 |
| GPT-4o | 68.8% | 7.14 | 7.97 | 5.84 | 6.96 |
| FakeShield [33] | 52.8% | 5.91 | 6.35 | 4.67 | 5.55 |
| **ForgerySleuth** | **92.2%** | **9.10** | **9.60** | **9.89** | **9.45** |

Table 12: Forgery analysis results of ForgerySleuth and SoTA methods, evaluated using STS.

| Model | **Ours** | FakeShield | GPT-4o | QWen2 | QWen | LISA |
|---|---|---|---|---|---|---|
| all-MiniLM-L6-v2 | **0.926** | 0.662 | 0.725 | 0.595 | 0.551 | 0.313 |
| all-MiniLM-L12-v2 | **0.919** | 0.514 | 0.645 | 0.505 | 0.475 | 0.350 |
| all-mpnet-base-v2 | **0.961** | 0.695 | 0.724 | 0.635 | 0.546 | 0.401 |

shown in the FakeShield row. The notably lower accuracy stems from the low recall rates, where FakeShield fails to detect manipulations, leading to significant missed detections. To further analyze FakeShield's localization performance, we also report its performance on only the successfully detected manipulated images (*i.e.*, True Positive samples). As indicated in the row for only true positive samples, its performance improves significantly under this setting, but it still falls short compared to ForgerySleuth. However, since manipulation detection is performed without prior knowledge of whether an image is manipulated, the first setting (which considers all images) provides a more realistic and fair evaluation, while the second setting serves only as a supplementary analysis.

**Forgery Analysis Results.** We employ GPT-4 as an evaluator on the ForgeryAnalysis-Eval dataset to assess the quality of textual analysis and reasoning. The evaluation considers several key dimensions: correctness, relevance, and detail. The results are presented in Table 11 (corresponding to Figure 1 (b)). FakeShield exhibits low recall rates, further corroborating the previously discussed issue of missed detections. Additionally, its analysis falls behind ForgerySleuth in both accuracy and level of detail.

However, since GPT-4 itself is susceptible to hallucinations, this approach does not guarantee a fully objective and quantitative evaluation. To address this limitation, we incorporate Semantic Textual Similarity (STS) to measure the similarity between the generated text and the ground-truth text in ForgeryAnalysis-Eval. The results, shown in Table 12, further confirm that our ForgerySleuth consistently outperforms FakeShield.

**Qualitative Results.** We present the performance of ForgerySleuth and FakeShield in manipulation localization and forgery analysis, as shown in Figure 8. The results demonstrate that ForgerySleuth not only achieves higher accuracy in localizing manipulated regions, but also generates more detailed and precise analysis.

# D Experiment

## D.1 Experimental Implementation Details

**Implementation Details.** We employ LLaVA-7B-v1-1 [14] as the base multimodal LLM ($\mathcal{F}_m$) and use the ViT-H SAM [42] backbone for the vision encoder ($\mathcal{F}_v$). For training, we utilize 2 NVIDIA 80GB A800 GPUs, with training scripts optimized by DeepSpeed [62], which helps reduce memory usage and accelerate training. We use the AdamW [63] optimizer, setting the learning rate to $0.0002$ with no weight decay. The learning rate is scheduled using WarmupDecayLR, with 100 warmup iterations. The weights for the text generation loss $\lambda_{\text{txt}}$ and mask loss $\lambda_{\text{mask}}$ are both set to $1.0$, while the BCE loss $\lambda_{\text{bce}}$ and DICE loss $\lambda_{\text{dice}}$ are weighted at $1.0$ and $0.2$, respectively. The batch size per device is $4$, with gradient accumulation steps set to $4$.

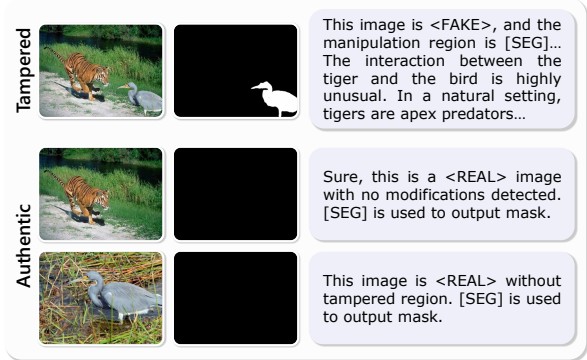

Figure 6: Predictions of ForgerySleuth on authentic and tampered images.

Table 13: Human evaluation results for the forgery analysis task.

| Method | Samples | *Recall* | Correctness | Relevance | Detail | Overall |
|--------|---------|----------|-------------|-----------|--------|---------|
| QWen2 | 139 | 35.3% | 4.95 | 5.10 | 2.85 | 4.11 |
| GPT-4o | 134 | 65.7% | 8.06 | 8.32 | 5.77 | 7.18 |
| **Ours** | 147 | **91.2%** | **9.48** | **9.53** | **9.61** | **9.44** |

## D.2 Manipulation Detection Results

Modeling the natural laws of real samples and distinguishing them from tampered images is crucial. We utilized two special tokens, <REAL> and <FAKE>, which indicate the detection results. In addition, the output mask provides valuable information for verifying image authenticity. Figure 6 illustrates the ForgerySleuth detection results for authentic and manipulated images, including the predicted tampering masks and textual analysis. Our method achieves real/fake classification accuracies of 0.989 on Columbia and 0.910 on CASIA1, demonstrating its effectiveness in manipulation detection.

## D.3 Forgery Analysis Results

We leverage GPT-4 for evaluation based on the ForgeryAnalysis-Eval dataset to assess the quality of text reasoning and explanations. Ratings are based on several dimensions, including the correctness of tampered objects, the relevance of clues to manipulation, and the detail of analysis, reflecting the capability of comprehension, reasoning, and correctness. We collect responses from each M-LLM, and GPT-4 assigns scores from 1 to 10 for each response, with higher scores indicating better performance. Beyond scoring, GPT-4 provides explanations for its ratings, ensuring transparency and consistency in the evaluation process. Figure 18 illustrates the prompt structure used for the evaluation and the response of GPT-4. To ensure consistent and fair scoring, the evaluation prompt includes clear scoring criteria for assessment aspects, including correctness, relevance, and details. GPT-4 assigns a score for each evaluation dimension and provides detailed comments to justify the rating.

To directly assess the ability to detect manipulated images, we explicitly instruct the evaluated M-LLMs through prompts to additionally output <FAKE> or <REAL> to indicate their detection results. All models, except LISA-7B-v1-explanatory, can provide the required response. We use the recall rate to reflect the ability to correctly identify tampered images.

**Human Evaluations.** We also conduct human evaluations under the same experimental setup as the evaluation using GPT-4 described above. A total of 14 volunteers participated, each scoring for 30 random samples. The results are summarized in Table 13.

## D.4 Qualitative Results

We present additional results of our ForgerySleuth in both forgery analysis and manipulation localization tasks. We also provide forgery analysis results from existing M-LLMs (GPT-4o and Qwen-VL)

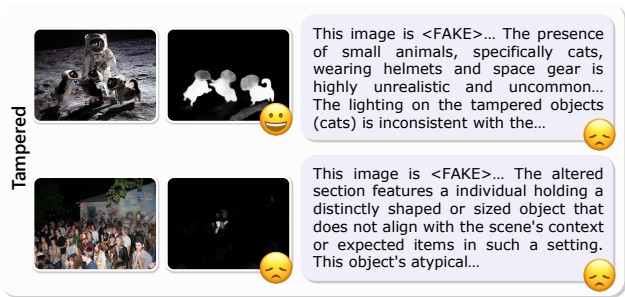

Figure 7: Failure cases in predictions of ForgerySleuth.

and segmentation masks from traditional image manipulation detection (IMD) methods, illustrating the advantages of our proposed IMD assistant in terms of accuracy and explainability. These cases are from public datasets IMD20 and NIST16, demonstrating the generalization capabilities of our method.

In the example in Figure 9, both M-LLMs classify the image as real without detecting manipulation. Similarly, most IMD methods, except CAT-Net2, fail to localize the manipulated regions accurately. However, ForgerySleuth identifies the tampered regions and provided a detailed tampering analysis. Figure 10 presents a case with more apparent tampering traces. GPT-4o and Qwen-VL both exhibit varying degrees of overthinking, leading to inaccurate analysis. Our method demonstrates higher localization precision and analytical accuracy. In the case of Figure 11, ForgerySleuth precisely localizes the tampered regions, even including the shadow of the person. Compared to the textual outputs of M-LLMs, the mask generated by ForgerySleuth provides a more intuitive and accurate detection result.

# E    Limitation and Future Work

## E.1    Failure Cases

We analyzed ForgerySleuth's detection failures and identified two representative error patterns.

**Semantic Description Inaccuracies.** In these cases, our model can accurately localize manipulated regions, but its textual analysis contains description errors. For instance, in the first example in Figure 7, the model misidentifies dogs as cats. Importantly, this does not impact the detection logic or localization accuracy. These errors often arise when specific semantic concepts in images are challenging to detect clearly. Future work could improve this by enhancing the visual understanding capabilities of MLLM and LLM backbone models.

**Failures with Small Manipulations in Complex Scenes.** In complex image content with small manipulated regions, the model's localization precision can degrade, sometimes leading to incorrect textual analysis and, consequently, a detection failure. For example, in the second instance in Figure 7, where a spliced cartoon character is present, the model only localized a partial manipulation (*e.g.*, the guitar in hand) while overlooking the character itself, primarily due to the small proportion of the manipulation and the image's overall complexity. Detecting small region manipulations remains a significant challenge, and MLLMs tend to focus on global semantic information rather than fine-grained local details. Therefore, developing methods to effectively detect anomalies in small regions is a crucial research direction.

## E.2    Constraints in Evaluation Methods

Our paper evaluates analysis accuracy using GPT-4o and STS methods on the ForgeryAnalysis-eval dataset. However, our evaluation methods are subject to specific constraints: our assessment relies on GPT-4o, a large external model that may exhibit inherent biases or hallucinations, and STS only reflects semantic similarity between two sentences. Thus, exploring more comprehensive and objective evaluation methods, including the quantitative assessment of model hallucinations, is an important direction for future work.

### E.3 Model Scale and Inference Costs

While leveraging LLMs to complement traditional image manipulation detection methods, which often primarily focus on low-level traces, this approach increases the model's scale and computational cost of generating detailed explanations. In the future, we plan to investigate model light-weighting solutions. This includes developing a version where generating such detailed output is optional, which would significantly reduce inference time. Alternatively, we could explore distilling LLM capabilities into smaller models, retaining only the essential knowledge for manipulation detection.

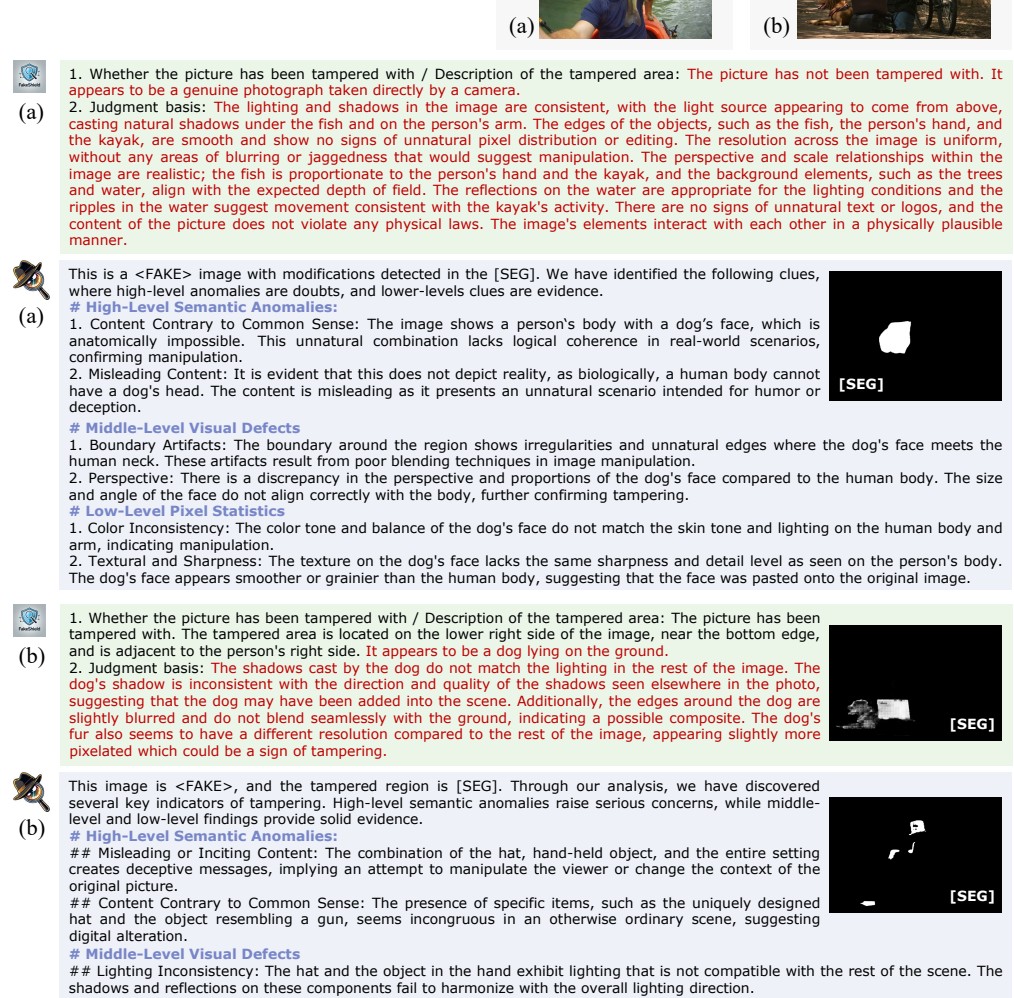

**(a)**
1. Whether the picture has been tampered with / Description of the tampered area: The picture has not been tampered with. It appears to be a genuine photograph taken directly by a camera.
2. Judgment basis: The lighting and shadows in the image are consistent, with the light source appearing to come from above, casting natural shadows under the fish and on the person's arm. The edges of the objects, such as the fish, the person's hand, and the kayak, are smooth and show no signs of unnatural pixel distribution or editing. The resolution across the image is uniform, without any areas of blurring or jaggedness that would suggest manipulation. The perspective and scale relationships within the image are realistic; the fish is proportionate to the person's hand and the kayak, and the background elements, such as the trees and water, align with the expected depth of field. The reflections on the water are appropriate for the lighting conditions and the ripples in the water suggest movement consistent with the kayak's activity. There are no signs of unnatural text or logos, and the content of the picture does not violate any physical laws. The image's elements interact with each other in a physically plausible manner.

**(a)**
This is a <FAKE> image with modifications detected in the [SEG]. We have identified the following clues, where high-level anomalies are doubts, and lower-levels clues are evidence.
**# High-Level Semantic Anomalies:**
1. Content Contrary to Common Sense: The image shows a person's body with a dog's face, which is anatomically impossible. This unnatural combination lacks logical coherence in real-world scenarios, confirming manipulation.
2. Misleading Content: It is evident that this does not depict reality, as biologically, a human body cannot have a dog's head. The content is misleading as it presents an unnatural scenario intended for humor or deception.
**# Middle-Level Visual Defects**
1. Boundary Artifacts: The boundary around the region shows irregularities and unnatural edges where the dog's face meets the human neck. These artifacts result from poor blending techniques in image manipulation.
2. Perspective: There is a discrepancy in the perspective and proportions of the dog's face compared to the human body. The size and angle of the face do not align correctly with the body, further confirming tampering.
**# Low-Level Pixel Statistics**
1. Color Inconsistency: The color tone and balance of the dog's face do not match the skin tone and lighting on the human body and arm, indicating manipulation.
2. Textural and Sharpness: The texture on the dog's face lacks the same sharpness and detail level as seen on the person's body. The dog's face appears smoother or grainier than the human body, suggesting that the face was pasted onto the original image.

**(b)**
1. Whether the picture has been tampered with / Description of the tampered area: The picture has been tampered with. The tampered area is located on the lower right side of the image, near the bottom edge, and is adjacent to the person's right side. It appears to be a dog lying on the ground.
2. Judgment basis: The shadows cast by the dog do not match the lighting in the rest of the image. The dog's shadow is inconsistent with the direction and quality of the shadows seen elsewhere in the photo, suggesting that the dog may have been added into the scene. Additionally, the edges around the dog are slightly blurred and do not blend seamlessly with the ground, indicating a possible composite. The dog's fur also seems to have a different resolution compared to the rest of the image, appearing slightly more pixelated which could be a sign of tampering.

**(b)**
This image is <FAKE>, and the tampered region is [SEG]. Through our analysis, we have discovered several key indicators of tampering. High-level semantic anomalies raise serious concerns, while middle-level and low-level findings provide solid evidence.
**# High-Level Semantic Anomalies:**
## Misleading or Inciting Content: The combination of the hat, hand-held object, and the entire setting creates deceptive messages, implying an attempt to manipulate the viewer or change the context of the original picture.
## Content Contrary to Common Sense: The presence of specific items, such as the uniquely designed hat and the object resembling a gun, seems incongruous in an otherwise ordinary scene, suggesting digital alteration.
**# Middle-Level Visual Defects**
## Lighting Inconsistency: The hat and the object in the hand exhibit lighting that is not compatible with the rest of the scene. The shadows and reflections on these components fail to harmonize with the overall lighting direction.
## Perspective Relationships: The placement and dimensions of the hat and the hand-held object do not align with the scene's perspective. These items appear hurriedly inserted, disregarding the scene's natural depth and scale.
## Traces of Tampered Region: Discernible lines mark the altered areas. For example, the edges of the hat and the object being held are unnaturally crisp and fail to blend smoothly with their surroundings.
**# Low-Level Pixel Statistics**
## Noise Inconsistency: The sections with the gun-like object, the hat, and the item near the dog reveal noise pattern discrepancies when compared to adjacent regions, indicating tampering.
## Color Discrepancy: The shades in the altered areas slightly differ from their surroundings, particularly evident in the hat and the item held by the individual.

Figure 8: Comparison of results between ForgerySleuth and FakeShield.

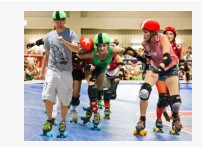 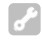

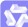 The image appears to be real, depicting a scene from a roller derby event. The participants are wearing protective gear such as helmets, knee pads, and elbow pads, which is typical for the sport. The expressions and body language of the individuals suggest they are actively engaged in the game. There are no obvious signs of digital manipulation, such as inconsistent lighting, shadows, or distorted proportions that would indicate the image has been altered. Additionally, the background shows spectators watching the event, adding to the authenticity of the scene.

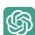 The image appears to be authentic based on several factors:
1. **Lighting and Shadows**: The lighting is consistent throughout the image. Shadows and highlights align with the assumed light sources.
2. **Proportions and Perspective**: The subjects' proportions and perspective match the setting. There are no noticeable inconsistencies with size or angle that often indicate manipulation.
3. **Blending and Edges**: The edges of the subjects are clean and well-defined with no visible signs of cut-and-paste operations, such as blurred or overly sharp outlines.
4. **Environment and Interaction**: The people in the scene interact naturally with each other and their surroundings. Details like the roller skates' contact with the floor and the interaction between individuals are logical and cohesive.
Overall, the elements in the image work together harmoniously, suggesting it is not manipulated.

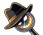 This is a <FAKE> image with modifications detected in the [SEG]. We have found several clues, where high-level anomalies create substantial doubts, and other-level findings contribute reliable evidence. 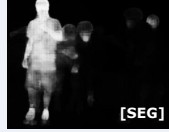

# High-Level Semantic Anomalies
1. **Content Contrary to Common Sense**:
 - The presence of a man in casual clothes amidst a group of roller derby players seems unusual, which raises suspicion. It is uncommon for an amateur player to be seen in the middle of the action along with the players.

# Middle-Level Visual Defects
1. **Lighting Inconsistency**:
 - The lighting on the tampered figure does not align with the lighting conditions of the surrounding environment. For instance, the shadows and highlights on the amateur player do not match the expected lighting direction and intensity observed on the other roller derby participants.
2. **Perspective Relationships**:
 - The perspective of the amateur player is slightly misaligned with the rest of the group. The positioning and relative size of the amateur player do not match naturally with the surrounding players, indicating that the figure was not part of the original scene.
3. **Physical Constraints**:
 - The amateur player's interaction with the other players is awkward. There is a lack of natural contact or engagement between the referee and the surrounding skaters, suggesting that the amateur player was added without considering the physical dynamics of the scene.

# Low-Level Pixel Statistics
1. **Noise Inconsistency**:
 - The tampered region shows a different noise pattern compared to the surrounding areas. The amateur player has a smoother texture that does not match the natural noise present in the background.
2. **Color Discrepancies**:
 - There are subtle differences in color grading between the tampered figure and the rest of the image. The amateur player has skin tones and lighting that do not perfectly match the other elements in the scene.
3. **Sharpness Variations**:
 - The edges of the amateur player are unnaturally sharp compared to the rest of the image. This disparity in sharpness indicates that the figure was cut and pasted from another source.

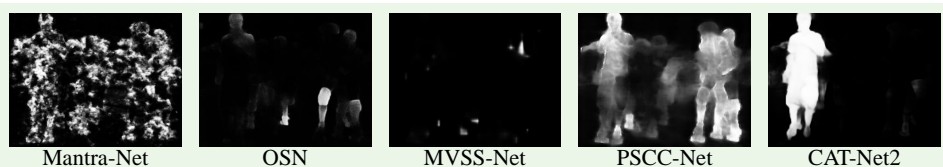

| Mantra-Net | OSN | MVSS-Net | PSCC-Net | CAT-Net2 |

Figure 9: Comparison of results between our ForgerySleuth, existing multimodal large language models, and traditional image manipulation detection methods. The image is c93gg4y_0.png from the IMD20 dataset.

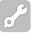 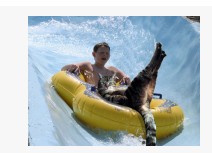

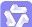 The image appears to be manipulated due to several inconsistencies:
1. **Cat's Position**: The cat is positioned in an unnatural way, with its legs extended upwards as if it were floating or being thrown into the air. Cats typically do not react like this when placed in water.
2. **Water Splash**: The splash of water around the tube does not seem consistent with the position of the boy and the cat.
3. **Lighting and Shadows**: The lighting and shadows on the cat do not match those on the boy and the tube. This suggests that the cat may have been added to the image after the original photo was taken.
4. **Edges and Blurring**: There are some areas where the edges of the cat appear slightly blurred or out of focus compared to the rest of the image, which can be a common indicator of digital manipulation.
These factors combined suggest that the image has been altered, most likely by adding the cat to the scene.

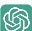 # Analysis of Manipulation Detection:
1. **Semantic Inconsistency**:
   - The scene depicts a child and a cat together on a water slide, with the cat in an extremely unnatural posture (especially with its leg raised high). This is highly unlikely in a real scenario, particularly given that cats generally avoid water.
   - While the inflatable tube's yellow surface blends reasonably well with the background, its style and lighting characteristics still seem slightly out of place, suggesting potential manipulation.
2. **Lighting and Physical Consistency**:
   - **Cat**: The lighting on the cat's body, particularly its raised leg, does not fully align with the bright environment of the water slide. The cat lacks strong reflective highlights that are visible on other objects.
   - **Inflatable Tube**: The tube's edge lighting appears slightly over-softened, and its sharpness compared to the background suggests potential blending issues.
3. **Edge Transition**:
   - Examining the contact between the cat and the inflatable tube reveals blending artifacts, particularly where the cat's hind leg meets the tube. The edges appear overly smoothed, hinting at post-processing.
   - Similarly, the inflatable tube's edge where it meets the water slide shows signs of digital manipulation, such as inconsistent reflections and mismatched textures.
4. **Behavioral Logic**:
   - It is highly implausible for a cat to participate in such an activity without showing fear or discomfort. The calm demeanor of the cat and the pose seem unnatural for this context.
Conclusion: Based on the above observations, both the cat and the inflatable tube exhibit signs of manipulation. The image is ultimately judged as **composite and manipulated (FAKE)**.

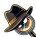 This is a <FAKE> image with modifications detected in the [SEG]. After a thorough examination, we have identified the following clues. High-level anomalies raise significant doubts requiring attention, while middle-level and low-level findings provide solid evidence. 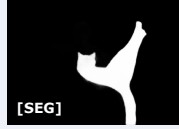
# High-Level Semantic Anomalies
1. **Common Sense Content**: The scenario depicted is highly unlikely. A cat calmly sitting in a flotation device on a water slide is unrealistic behavior, indicating manipulation.
2. **Misleading Content**: This image presents a false scenario intended to evoke humor or amusement, which is inherently misleading.
# Middle-Level Visual Defects
1. **Lighting Inconsistency**: The lighting on the tampered region does not align with the rest of the image. Reflections on the water and the boy's body are inconsistent with the lighting direction and intensity.
2. **Perspective Relationships**: The positioning and perspective of the cat in the flotation device do not naturally align with a human occupant. The cat appears unnaturally positioned, lacking the expected interaction with the water and surroundings.
3. **Physical Constraints**: The size and position of the cat relative to the boy indicate tampering. The cat is unnaturally large compared to the boy and is placed in an awkward position that defies physical realism.
# Low-Level Pixel Statistics
1. **Noise Inconsistency**: The noise pattern in the tampered region differs from the surrounding areas. The texture of the water and the flotation device around the cat exhibit different levels of graininess and noise, indicating manipulation.
2. **Color Discrepancy**: The color tones in the tampered region, particularly the cat, do not blend seamlessly with the surrounding water and the boy's skin. This suggests that the cat was added from a different source with different lighting conditions.
3. **Sharpness Variations**: The edges of the tampered region, particularly around the cat's body, appear unnaturally sharp compared to the rest of the image, indicating a cut-and-paste operation.

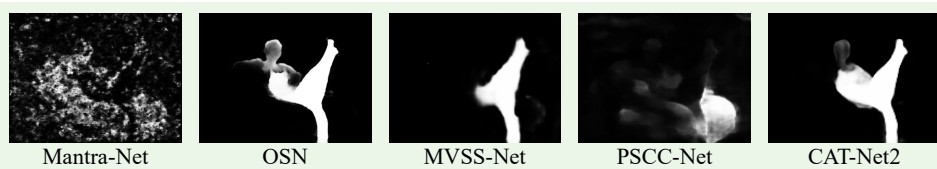

| Mantra-Net | OSN | MVSS-Net | PSCC-Net | CAT-Net2 |

Figure 10: Comparison of results between our ForgerySleuth, existing multimodal large language models, and traditional image manipulation detection methods. The image is c8zf0hg_0.jpg from the IMD20 dataset.

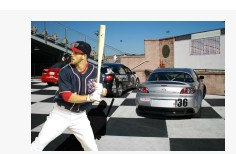

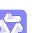 The image appears to be manipulated for several reasons:
1. **Foreground Subject**: The person holding the baseball bat is not naturally integrated into the scene. The lighting, shadows, and overall quality of the person do not match the background. The edges around the person, especially at the bottom, appear to have been cut out digitally.
2. **Background Context**: The setting looks like a race track or a car show, which is an unusual place for someone to be playing baseball. This mismatch in context suggests that the person has been added to the image artificially.
3. **Lighting and Shadows**: The lighting on the person does not align with the lighting in the rest of the image. The shadows cast by the cars and other objects in the background do not correspond with the position of the person.
4. **Image Quality**: There is a noticeable difference in the resolution and clarity between the person and the background. The person's image is sharper and more detailed compared to the slightly blurred background.
These factors collectively indicate that the image has been manipulated.

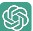 <FAKE>
Analysis:
1. **Lighting and Shadows**: The shadow of the person holding the bat does not align well with the other shadows in the scene, indicating a possible manipulation. The shadow is also very sharp and dark compared to the rest of the scene, which suggests inconsistency.
2. **Alignment and Perspective**: The person with the bat does not appear to be integrated smoothly into the environment. The size and angle of the figure seem off compared to the surrounding elements, like the cars.
3. **Integration with Background**: The person holding the bat appears to be layered on top of the existing background. The edge around the figure looks too clean and sharp, lacking the natural blending seen in real-world images.
4. **Context Clarity**: A person in a baseball uniform standing in a car racing environment seems out of context, adding to the suspicion of image manipulation.

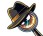 This is a <FAKE> image with modifications detected in the [SEG]. We have found several clues, where high-level anomalies create substantial doubts, and other-level findings contribute reliable evidence.

# High-Level Semantic Anomalies

1. **Content Contrary to Common Sense**: The presence of a professional baseball player in a full uniform, standing in a parking lot next to sports cars, is contextually odd. Such a scenario is improbable, suggesting that the image was staged or the player was placed into this environment unnaturally.
2. **Misleading Content**: The image may give the impression that the athlete is involved in or supports a particular car brand, which could be considered advertorial content without clear disclosure.

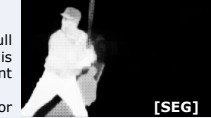

# Middle-Level Visual Defects

1. **Lighting Inconsistency**: The lighting on the athlete does not align with the lighting direction and intensity of the cars and background. Shadows and highlights on the athlete do not correspond naturally with the sunlight and shadows cast by the cars, indicating manipulation.
2. **Boundary Traces**: Noticeable edges where the athlete meets the background indicate possible cut-and-paste manipulation. These boundary artifacts are not smooth and visible, suggesting improper blending.
3. **Perspective Relationships**: The athlete is standing in a manner that does not fit naturally within the scene's perspective. The athlete's position and the angle are not aligned with the cars and the background, indicating that the image of the athlete was inserted from a different source.

# Low-Level Pixel Statistics

1. **Noise Inconsistency**: The noise pattern in the tampered region differs from that of the surrounding background. The athlete's section has a different grain or noise texture than the rest of the image, suggesting manipulation.
2. **Color Discrepancy**: The color tones and saturation levels of the athlete do not match the surrounding environment. The lighting on the athlete's uniform does not correspond with the overall lighting direction of the scene, indicating tampering.
3. **Sharpness/Blurriness**: There is a noticeable difference in sharpness between the tampered region and the background. The edges around the athlete are too sharp compared to the rest of the image, suggesting manipulation.

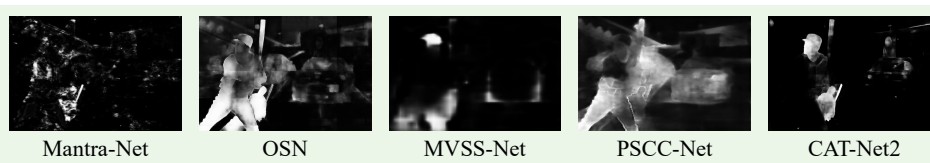

| Mantra-Net | OSN | MVSS-Net | PSCC-Net | CAT-Net2 |

Figure 11: Comparison of results between our ForgerySleuth, existing multimodal large language models, and traditional image manipulation detection methods. The image is NC2016_7312.jpg from the NIST16 dataset.

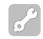

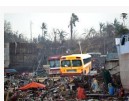 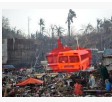

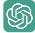

This is a FAKE image manipulated using AI-generated content, specifically targeting the bus in the center of the image, which is highlighted in red in the reference. Below is a breakdown of the detection clues across various levels:

Low-Level Pixel Statistics:
- Noise and Texture Discrepancies: The overall scene, which depicts a post-disaster landscape, contains natural graininess, subtle noise, and rough textures consistent with debris and rubble. However, the bus in the middle shows different characteristics, it appears smoother and lacks the same level of noise as its surroundings, indicating that it was generated or inserted separately. This noise discrepancy is a common marker of AI-generated content.
- Color and Sharpness Anomalies: The bus has unnaturally vivid colors (yellow and black), which stand out in contrast to the muted tones of the devastated environment. The sharpness of the bus also does not match the slightly blurred background, implying that it was not part of the original scene. This sharpness inconsistency indicates a different image source or digital generation.

Middle-Level Visual Defects:
- Boundary Traces: The edges of the bus appear unnaturally smooth, especially around the rear end and where it blends into the background. This smoothness is in stark contrast to the jagged debris and uneven structures in the environment. The lack of blending or transition between the bus and its surroundings, especially along its base, highlights its insertion into the scene.
- Lighting Inconsistency: The overall lighting in the image is diffused and soft, typical of overcast post-disaster conditions. The bus, however, shows inconsistent lighting, with reflective surfaces that suggest stronger light sources that do not exist in the surrounding scene. The shadows on the bus are less pronounced and don't match the diffused light in the image, further indicating manipulation.
- Perspective and Physical Constraints: The bus appears floating or elevated, which defies the physical laws of gravity and common sense. There is no clear indication of support or ground contact for the bus, suggesting it was artificially placed in this environment. Moreover, the size and angle of the bus do not match the perspective of the background elements, further reinforcing the hypothesis that it was generated and inserted unnaturally.

High-Level Semantic Anomalies:
- Contrary to Common Sense: From a high-level contextual perspective, the bus being suspended in such a chaotic, disaster-stricken environment is illogical. Buses do not float mid-air, and there are no physical cues in the scene (such as debris lifting or supporting the bus) to explain its position. This is a direct indication of content manipulation.
- Inappropriate Object Placement: In a disaster scenario like this, objects like vehicles are usually scattered, damaged, or buried within the debris. The bus, however, appears undamaged and disproportionately out of place. The placement of the bus in this specific location within the scene is highly unnatural and lacks coherence with the overall narrative of destruction.

Conclusion:
This image has been manipulated using AI-generated content, with the tampered region being the floating bus in the middle. The analysis of noise patterns, boundary traces, lighting inconsistencies, and semantic illogic clearly indicate that the bus was artificially inserted into the scene. This manipulation disrupts the overall realism of the disaster environment and highlights the telltale signs of AI-generated imagery.

This image is **<FAKE>**, and the tampered region is **[SEG]**. We have identified the following clues, where the high-level anomalies are significant doubts worth attention, and the middle-level and low-level findings are reliable evidence.

**# High-Level Semantic Anomalies**
- Contrary to Common Sense: From a high-level contextual perspective, the bus being suspended in such a chaotic, disaster-stricken environment is illogical. Buses do not float mid-air, and there are no physical cues in the scene (such as debris lifting or supporting the bus) to explain its position.
- Inappropriate Object Placement: In a disaster scenario like this, objects like vehicles are usually scattered, damaged, or buried within the debris. The bus, however, appears undamaged and disproportionately out of place.

**# Middle-Level Visual Defects**
- Lighting Inconsistency: The overall lighting in the image is diffused and soft, typical of overcast post-disaster conditions. The bus, however, shows inconsistent lighting, with reflective surfaces that suggest stronger light sources that do not exist in the surrounding scene. The shadows on the bus are less pronounced and don't match the diffused light in the image.
- Perspective and Physical Constraints: The bus appears floating or elevated, which defies the physical laws of gravity and common sense. There is no clear indication of support or ground contact for the bus, suggesting it was artificially placed in this environment. Moreover, the angle of the bus does not match the perspective of the background elements.

**# Low-Level Pixel Statistics**
- Noise and Texture Discrepancies: The overall scene contains natural graininess, subtle noise, and rough textures. However, the bus in the middle shows different characteristics. It appears smoother and lacks the same level of noise as its surroundings. This noise discrepancy is a common marker of AI-generated content.
- Color Anomalies: The bus has unnaturally vivid colors, which stand out in contrast to the muted tones of the devastated environment.
- Sharpness Anomalies: The sharpness of the bus also does not match the slightly blurred background, implying that it was not part of the original scene. This sharpness inconsistency indicates a different image source or digital generation.

Figure 12: Illustration of the ForgeryAnalysis dataset creation process. The data is initially generated by GPT-4o, then revised by experts to ensure the accuracy of the analysis, with clues organized in the Chain-of-Clues (CoC) format.

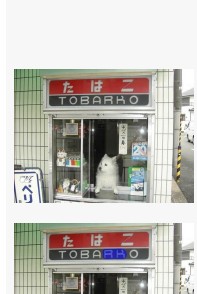

# High-Level Semantic Anomalies:

1. **Content Contrary to Common Sense**:
 - **The phrase "UNDER 20 IN DOG YEARS" in the lower window sign is semantically unusual and unlikely to be displayed in such a context.** Typically, age restrictions are set in human years, highlighting the fakeness of the alteration.

# Middle-Level Visual Defects:

1. **Traces of Tampered Regions or Boundaries**:
 - The tampered letters have borders that look slightly abrupt against the background, revealing the typical telltale signs of cut-and-paste operations. The edges around the tampered text appear mismatched and inconsistent.
2. **Lighting Inconsistency**:
 - The lighting on the tampered area of the sign in the window's lower right corner ("UNDER 20 IN DOG YEARS") does not reflect the lighting conditions of the surrounding environment. The natural reflections and shadows do not match the rest of the image.
3. **Perspective Relationships**:
 - There is a slight perspective inconsistency in the alignment of the tampered letters "R" and "K". The letters do not perfectly follow the perspective lines of the original text's plane.

# Low-Level Pixel Statistics:

1. **Noise and Texture Inconsistency**:
 - The tampered text "TOBARKO" shows inconsistencies in noise patterns and textural details when compared with the rest of the image. The non-edited parts have a finer texture, while the "R" and "K" characters in "TOBARKO" show slightly different graininess.
2. **Color Discrepancies**:
 - There is a subtle mismatch in the color tones of the tampered letters compared to the original characters. The tampered letters display a less natural integration with the background color gradient.
3. **Sharpness and Blurring**:
 - The tampered areas, especially the letters "R" and "K", exhibit different sharpness levels compared to their surroundings. This indicates they have been copied and pasted from a different source.

CASE 1. Semantically abnormal text.

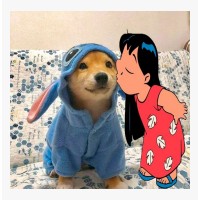
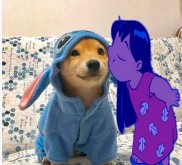

# **High-Level Semantic Anomalies:**

 - **Uncommon Content:** **The presence of a cartoon character interacting with real-world elements is semantically odd and unlikely**, alerting viewers to potential manipulation.
 - **Contradiction to Physical Reality:** The cartoon character, by nature, does not adhere to real-world physical constraints. Its existence and interaction within the scene defy common logical and physical expectations, such as shadow casting and 3D interaction with the environment.

# **Middle-Level Visual Defects:**

 - **Boundary Traces:** The boundaries around the cartoon character are not seamless. Upon close inspection, there can be visible traces of tampering work, such as edge haloing or pixel misalignment.
 - **Lighting Inconsistency:** The lighting on the cartoon character does not match the lighting conditions of the actual scene. The cartoon's illumination seems uniform, while the real-world elements show natural lighting variations and shadows.
 - **Perspective Issues:** The cartoon character is visually inconsistent in perspective alignment with the rest of the scene. The size, posture, and depth are not coherent with the surrounding real objects, making it evident that it was digitally inserted.

# **Low-Level Pixel Statistics:**

 - **Noise Inconsistency:** The tampered region exhibits a different noise pattern compared to the rest of the image. For example, the cartoon character's area has smoother textures, whereas the surrounding image has more pronounced noise granularity.
 - **Color Anomalies:** The color blending around the intersection of the cartoon character and the real-world objects shows inconsistencies. Specifically, the transition at the boundary is not smooth and does not match the surrounding color palette.
 - **Sharpness Disparities:** The boundary between the natural image elements and the inserted cartoon character exhibits differences in edge sharpness. The cartoon edges are unnaturally sharp compared to the organic nature of the rest of the scene.

CASE 2. Presence of elements with different styles.

Figure 13: Examples from the ForgeryAnalysis-Eval dataset. The data is initially generated by GPT-4o and then cross-revised by multiple experts. The manipulation type for these images is "splice".

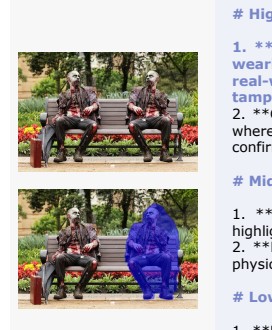

**# High-Level Semantic Anomalies:**

1. **Duplicated Content:** The presence of two identical figures sitting in the exact same position, wearing the same attire, and having identical poses and damages is highly unnatural and contrary to real-world circumstances. This resemblance is typically an indicator of image cloning or copy-paste tampering.
2. **Common Sense Violation:** The scenario depicted breaks the general rules of common sense and realism, where identical individuals appearing as zombies and mirroring each other's positions is highly improbable, confirming suspicions of artificial manipulation.

**# Middle-Level Visual Defects:**

1. **Lighting Inconsistencies:** There is a noticeable difference in shadow and lighting. The shadows and highlights on the manipulated person (on the right side) reflect slight inconsistencies compared to the image's rest.
2. **Perspective Discrepancy:** The perspective of the manipulated figure does not align perfectly with natural physics.

**# Low-Level Pixel Statistics:**

1. **Noise Inconsistency:** Upon close inspection, the noise pattern in the tampered region is slightly different from the rest of the image.

CASE 3. Duplicated content.

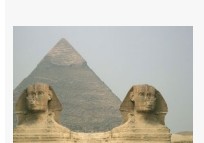

**# High-Level Semantic Anomalies**

1. **Content Contrary to Common Sense**:
 - Knowing the historical and cultural context of the Great Sphinx, having two Sphinxes side-by-side is historically inaccurate. This anomaly contradicts common historical knowledge.
 - Semantic understanding of heritage sites indicates that only one Great Sphinx exists, making the duplicated Sphinx a blatant anomaly.

**# Middle-Level Visual Defects**

1. **Lighting Inconsistencies**:
 - The right Sphinx reveals subtle differences in shadow and lighting compared to the copied region, hinting at the inconsistency.

2. **Perspective Relationships**:
 - The proportions and alignment between the pyramid and the tampered Sphinx area exhibit perspective mismatches.

**# Low-Level Pixel Statistics**

1. **Noise Patterns**:
 - Upon close inspection, the tampered region exhibits a different noise level or pattern compared to the rest of the image.

2. **Textural Mismatches**:
 - Fine details such as erosion patterns on the Sphinx show unnaturally replication.

CASE 4. Contradict common historical knowledge.

Figure 14: Examples from the ForgeryAnalysis-Eval dataset. The data is initially generated by GPT-4o and then cross-revised by multiple experts. The manipulation type for these images is "copy-move".

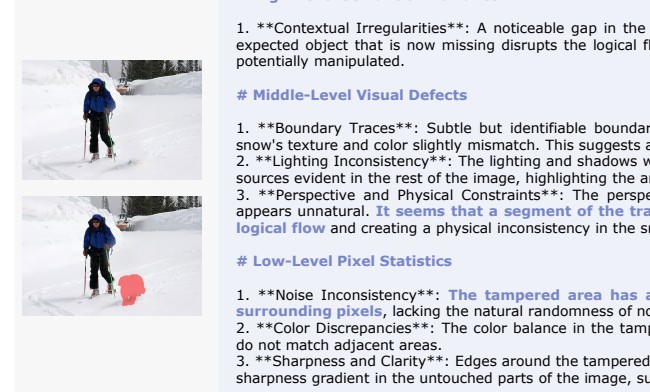

# High-Level Semantic Anomalies

1. **Contextual Irregularities**: A noticeable gap in the snow where tracks should have been continuous or an expected object that is now missing disrupts the logical flow of the scene. This suggests something unusual and potentially manipulated.

# Middle-Level Visual Defects

1. **Boundary Traces**: Subtle but identifiable boundaries are present around the tampered area, where the snow's texture and color slightly mismatch. This suggests a poorly executed removal of an object or person.
2. **Lighting Inconsistency**: The lighting and shadows within the tampered region do not align with natural light sources evident in the rest of the image, highlighting the artificial modification.
3. **Perspective and Physical Constraints**: The perspective of the snow tracks around the tampered region appears unnatural. **It seems that a segment of the tracks has been removed or modified, disrupting the logical flow** and creating a physical inconsistency in the snow-covered environment.

# Low-Level Pixel Statistics

1. **Noise Inconsistency**: **The tampered area has a noticeably smoother noise pattern compared to surrounding pixels**, lacking the natural randomness of noise seen in the rest of the image.
2. **Color Discrepancies**: The color balance in the tampered region appears off, with slight shifts in tones that do not match adjacent areas.
3. **Sharpness and Clarity**: Edges around the tampered region show unusual blurriness compared to the natural sharpness gradient in the untouched parts of the image, suggesting imperfect integration.

CASE 5. Noise inconsistency.

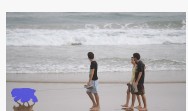

# High-Level Semantic Anomalies

1. **Content Contrary to Common Sense**:
 - **The presence of a leash and its shadow suggests a pet (such as a dog) should be visible**, but it is missing, which contradicts common sense and indicates manipulation.

# Middle-Level Visual Defects

1. **Traces of Tampered Region/Boundary**:
 - The boundary of the tampered area shows unnatural lines and edges, indicating manipulation. These traces suggest an object was removed without seamlessly blending the surrounding pixels.
2. **Lighting Inconsistency**:
 - The lighting in the manipulated region, including the reflection, does not match the direction and intensity of the light in the rest of the scene, revealing tampering.
3. **Perspective Relationships**:
 - **The reflection in the water suggests that an object was there**, but the size and shape of the reflection do not align with the surrounding reflections, indicating image editing.

# Low-Level Pixel Statistics

1. **Noise Distribution**:
 - The noise pattern in the tampered region is inconsistent with the rest of the image, appearing more smoothed, which indicates manipulation.
2. **Color Discontinuity**:
 - There is a color mismatch between the tampered region and its surroundings, with the tones and hues in the reflection not perfectly matching the water and beach.
3. **Sharpness and Blurriness**:
 - The boundaries of the tampered region are either unnaturally sharp compared to the rest of the image, suggesting poor blending during the editing process.

CASE 6. Unusual shadows and reflections.

Figure 15: Examples from the ForgeryAnalysis-Eval dataset. The data is initially generated by GPT-4o and then cross-revised by multiple experts. The manipulation type for these images is "remove".

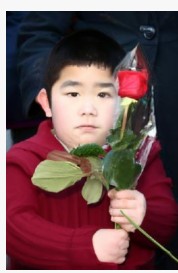

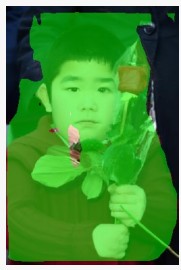

# High-Level Semantic Anomalies

1. **Content and Common Sense**:
 - The flower's appearance and positioning are slightly off, given the natural posture and handling by the young child.

# Middle-Level Visual Defects

1. **Traces of Tampered Region or Boundary**:
 - The boundary of the manipulated area appears overly distinct and does not merge well with the natural contextual edges of other original elements in the image. There are visible aberrations along the edges where the tampered section meets the natural photo.
2. **Lighting Inconsistency**:
 - The lighting on the manipulated rose does not match the natural lighting on the child's face and hands. The direction and intensity of shadowing on the tampered area are inconsistent with the rest of the scene, suggesting an addition that does not conform to the natural light sources in the original image.
3. **Physical Constraints**:
 - Upon close inspection, the way the child's fingers interact with the flower appears unnatural, with slight misalignment, further confirming that the flower has been digitally manipulated into the scene.

# Low-Level Pixel Statistics

1. **Noise Inconsistencies**:
 - While the background and the rest of the child's features have consistent noise distribution, the tampered part of the rose exhibits a smoother texture, indicating noise reduction typical in composite images.
2. **Color and Textural Mismatches**:
 - The color gradient and shading on the tampered part of the rose do not blend seamlessly with the surrounding real elements.
3. **Sharpness Discrepancies**:
 - The boundary of the tampered region is unusually sharp compared to its surroundings. The difference in sharpness is a strong low-level clue that suggests an overlay.

CASE 7. Unnatural interaction.

# High-Level Semantic Anomalies:

1. **Content Plausibility**: While somewhat plausible, the suspended plane in this context is unusual. Most cranes are not designed for such delicate operations, and the scene might evoke skepticism regarding its realness, raising suspicion of manipulation.
2. **Physical Constraints**: The suspension points and the way the plane is held appear physics-defying. The ropes and balance don't align with the expected real-world weight distribution and gravitational effects on such an object.

# Middle-Level Visual Defects:

1. **Lighting Inconsistency**: The lighting on the plane doesn't match the lighting on other objects and people in the scene. Specifically, look at the shadows on the plane in the marked region-they appear unnatural compared to how light interacts with the surrounding environment.
2. **Boundary Traces**: There are subtle boundary artifacts around the tampered plane region. The boundaries of the plane show a certain level of pixel interpolation which often occurs during copy-paste tampering, leading to visible seams upon close inspection.
3. **Perspective Relationships**: There is a slight perspective mismatch between the plane and the surrounding elements, including the rope and the ship. The alignment and proportions appear off, which indicates a manipulation.

# Low-Level Pixel Statistics:

1. **Noise Patterns**: The noise in the tampered region (the plane) is inconsistent with other parts of the image. Typically, authentic images have uniform noise distribution, while AI-generated manipulations often have differing noise characteristics in tampered parts.
2. **Color and Sharpness**: The colors in the tampered region are slightly more saturated, and the edges are sharper compared to the rest of the image. Authentic objects tend to blend naturally with the ambient softness and color consistency of the picture.

CASE 8. Physics defying.

Figure 16: Examples from the ForgeryAnalysis-Eval dataset. The data is initially generated by GPT-4o and then cross-revised by multiple experts. The manipulation type for these images is "AI-generate".

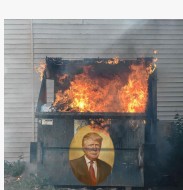

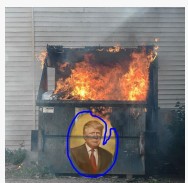

# High-Level Semantic Anomalies

1. **Content Contrary to Common Sense**:
 - The presence of a prominent, well-known individual on a trash fire is highly unusual and unlikely in real-world contexts, raising immediate suspicion about the image's authenticity.
2. **Inciting and Misleading Content**:
 - The juxtaposition of a public figure with such an explosive and destructive background can be misleading and potentially inciting. This unusual context strongly suggests an intent to provoke a reaction or convey a false narrative.

# Middle-Level Visual Defects

1. **Boundary Traces**:
 - The edges around the tampered region are irregular and show unnatural transitions. Evidence of this includes slight color and texture mismatches along the perimeter of the face.
2. **Lighting Inconsistency**:
 - The lighting on the face does not match the rest of the scene. The smoke and flames surrounding the face should exhibit the same lighting effects (reflection, shadow, glow); however, the tampered region shows inconsistent lighting, which is a strong indicator of manipulation.
3. **Perspective Relationships**:
 - The perspective of the face does not align well with the rest of the object ( trash can and flames ). It seems as if the face has been pasted onto the image without considering the overall perspective, making it look out of place.

# Low-Level Pixel Statistics

1. **Noise and Texture**:
 - The tampered region (the face) exhibits different noise patterns compared to the rest of the image. The texture of the face is smoother, while the surrounding smoke and flames have more granular noise, indicating manipulation.
2. **Color and Sharpness**:
 - The tones and sharpness of the face do not match the rest of the scene. The face is well-defined, while the surrounding elements appear slightly blurred due to the smoke and fire.

# High-Level Semantic Anomalies

1. **Content Contrary to Common Sense**:
 - The presence of a person in a roller coaster seat without any visible means of support is contrary to common amusement park rides, where riders are typically seated in enclosed cars or restrained in some manner.
 - The absence of such restraints, as well as the lack of seat belt or barrier in the vehicle, makes the scenario implausible.

# Middle-Level Visual Defects

1. **Lighting Inconsistency**:
 - The lighting on the person in the tampered region does not match the lighting conditions of the rest of the scene. For instance, the roller coaster lacks shadows or reflections that should correspond to the person if they were genuinely there.
2. **Traces of Tampered Region or Boundary**:
 - Upon close inspection, the boundaries of the tampered area reveal subtle artifacts such as unnatural edges or blending issues, suggesting it was superimposed onto the scene.

# Low-Level Pixel Statistics

1. **Noise Analysis**:
 - The tampered region shows a different noise pattern compared to the rest of the image. The natural noise in the roller coaster seat should be consistent, but the tampered area displays irregular noise levels that differ from the background.
2. **Color Consistency**:
 - There are slight color mismatches between the person and the background. The tampered region has a different hue and saturation, which suggests that the person was added from a different source image with different lighting conditions.
3. **Sharpness**:
 - The region and edges around the person are unnaturally sharp compared to the rest of the image, indicating they were superimposed onto the scene.

Figure 17: Examples from the ForgeryAnalysis-PT dataset. The data is automatically generated by our data engine ForgeryAnalyst.

Given the tampered image, the reference forgery analysis text, and the generated analysis text, assess the quality of the generated analysis based on the following criteria. Please note! TAMPERED OBJECTS refer to "dog," "face," "grass," etc., while "boundary artifacts" or "lighting inconsistencies" serve as CLUES. 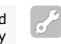

1. Correctness of Tampered Object (NOT Clues) Detection
- 8-10 points: ALL clues accurately identify major tampered objects and describe alterations but may miss subtle details.
- 4-7 points: ALL clues reveal the tampered regions but lack clarity or miss some regions.
- 1-3 points: Score in this range if ANY clues fail to identify tampered objects, or ANY incorrect detection is present.
2. Relevance of Clues to Tampered Object
- 8-10 points: ALL or MOST clues directly relate to tampered elements, logically supporting the analysis.
- 4-7 points: SOME clues are loosely related and lack a clear connection to the tampered content.
- 1-3 points: ALL Clues are irrelevant or misleading, not contributing to the tampering analysis.
3. Detail of Clues and Analysis
- 9-10 points: Provides comprehensive and well-explained clues with thorough reasoning.
- 6-8 points: Contains good detail but may lack depth in some aspects.
- 3-5 points: Limited detail with some clues inadequately explained.
- 1-2 points: Minimal detail with insufficient reasoning or vague explanations.

Assign separate scores on a scale of 1 to 10 for each criterion, with a higher score reflecting better performance. Provide the scores and concise comments explaining the assessment in the format: "Score1: {score}/10 \n Comment1: {text} \n Score2: {score}/10 \n Comment2: {text} \n Score3: {score}/10 \n Comment3: {text}"

**Reference Analysis Text:** {
We have identified the following clues, where high-level anomalies are doubts, and lower-levels clues are evidence.
# High-Level Semantic Anomalies: ## Content Contrary to Common Sense: The dog's face on a person's body is impossible and contrary to common sense. This unnatural combination lacks logical coherence in real-world scenarios, confirming manipulation. The content is misleading as it presents an unnatural scenario intended for humor or deception…
}

**Generated Analysis Text:** {
<FAKE> The image appears to be digitally manipulated. The most obvious clue is the fish being held by the person's hand. Fish do not have hands, so this element seems out of place and artificial. Additionally, the human's face looks like it has been edited onto the fish, which further suggests that the image was altered post-production. There are no visible signs of tampering at the pixel level, but the overall composition and elements present in the image indicate that it has been edited.
}

Score1: 1/10
Comment1: The generated analysis fails to correctly identify the tampered objects present in the reference analysis. Instead of recognizing the dog's face on the human body, it incorrectly discusses a fish being held, which is not mentioned in the reference text.

Score2: 1/10
Comment2: The clues provided in the generated analysis are irrelevant to the tampered object discussed in the reference analysis. The mention of a fish is misleading and not related to the actual tampered content, which involves a dog's face on a human body.

Score3: 2/10
Comment3: The detail in the generated analysis is minimal and not aligned with the actual tampering described in the reference text. The analysis lacks depth, particularly in explaining the clues, and incorrectly focuses on detecting non-existent elements, providing no useful insight into the actual manipulation.

Figure 18: Illustration of the prompt and GPT-4 response of text analysis quality evaluation. The evaluation prompt includes clear scoring criteria for the assessment aspects, ensuring consistent and fair scoring.

