# OpenReview forum: "ForgerySleuth: Empowering Multimodal Large Language Models for Image Manipulation Detection"
_NeurIPS.cc/2025/Conference — NeurIPS 2025 poster_

### Official Review · Reviewer_oWxz · 2025-06-03

**Clarity:** 2
**Significance:** 3
**Originality:** 2
**Rating:** 4
**Confidence:** 3

**Summary:**

The paper is introducing a dataset (ForgeryAnalysis) and a method (ForgerySleuth) for image manipulation detection. Both leverage MLLMs to generate text captions in the dataset and then train an MLLM for the classification and forgery mask prediction.

**Questions:**

There are several unclear areas in both the method and the experimental results presented, extensively analysed above. The main issues that would need to be addressed are clarity in the method and more detailed ablations.

**Ethical Concerns:**

["NO or VERY MINOR ethics concerns only"]

**Final Justification:**

Thank you for the very detailed responses to all reviews. Most of the concerns are addressed by the rebuttal; however, the discussion lacked transparency. Specifically, the questions asked originally were not answered directly or were evaded, raising questions about the understanding and interpretation of the paper. As such, I have raised the score, but I am less confident about it post-rebuttal.

**Limitations:**

yes

**Paper Formatting Concerns:**

The formatting is fine, but it does feel that some important methodological details and results are left in the appendix. Authors should consider restructuring and limiting  some sections (eg. Implementation details and metrics, related work) to bring important information in the main paper.

**Quality:**

3

**Strengths And Weaknesses:**

The paper addresses the need for task specific datasets and methods leveraging the common sense reasoning of MLLMs. Specifically, the introduction of a dataset with captions should encourage research in the task. Moreover, the results obtained by ForgerySleuth are good compared to previous works on the detection task.

Several aspects of the proposed method are unclear and are not properly explained in the paper.
- What prompt is used for GPT4? This should be shown in the text and figure 2
- Does GPT4 calculate the low, mid and high level "clues" or simply rephrase based on a pre-computed set of features?
- What kind of responses did GPT4 produce? Are there any examples of good vs hallucinated responses?
- For the CoC, it is stated that human experts curated the GPT4 responses. how many experts per sample and how is consistency ensured?
- No statistics of the introduced dataset are provided to better understand the composition of the dataset
- In terms of the ForgerySleuth training, it is unclear what the prompt x_p is.
- It is also unclear whether the CoC produced by GPT4 is used to train the model autoregressively or whether it is simply T
- Lines 182-183 mention enriching the LLM vocabulary with [SEG], <REAL> and <FAKE> tokens. Are these new embeddings or trainable tokens similar to the [CLS]? The phrasing and notation is ambiguous, as tin Fig. 3 it appears that embeddings are added but then a trainable token it taken for h_seg.
- Fusion mechanism seems like a key part of the method, but is left to the appendix.

In terms of the results, the localisation results seems convincing but more details on the detection and the qualitative output of the text should be included in the paper:
- Table 4: Llava is finetuned to produce part of the dataset. How does that perform on the STS?
- Classification (real/fake) performance should also be included as STS is not necessarily capturing this
- The ablation is incomplete: How does the classification/mask prediction perform without the CoC reasoning? how does the vision decoder perform without the LLM output?
- Are there any failure cases? Is there a pattern in the failure cases? The appendix includes 2 failure cases, this is not enough to draw conclusions.

---

> ### Author Rebuttal · Authors · 2025-07-31
>
> We sincerely appreciate your review and constructive suggestions. We acknowledge that some crucial details were initially placed in the Appendix, which might have led to confusion. Your valuable feedback has prompted us to integrate more key information directly into the main text.
>
> **Q1. More details on the dataset.**
>
> **Q1.1.** GPT-4o prompt details.
>
> The prompt of GPT-4o consists of two parts: first, the role and task instructions (detailed in **Appendix Lines 542-555**), and second, the specific per-sample input prompt (illustrated in **Appendix Figure 14**). Due to text length constraints, we can not place the full prompt in Figure 2, but we have added clear indicators in the main text to ensure readers can easily find these details in the Appendix.
>
> **Q1.2.** GPT-4o clues generation.
>
> We appreciate you highlighting this point. We have clarified in **Line 136** of the main text that:
>
> *We utilize the advanced M-LLM, GPT-4o, to generate the initial clue analyses based on its visual understanding of the input image (with highlighted regions) and its inherent world knowledge. For more detailed content and examples of GPT-4o response, please refer to Appendix Figure 14.*
>
> **Q1.3.** Examples of GPT-4o response.
>
> **Appendix Figure 14** and **Lines 556-565** illustrate the initial clues generated by GPT-4o and detail how human experts correct ambiguous or hallucinated parts. We've now explicitly labeled cases of both high-quality responses and instances that required correction due to hallucination or vagueness.
>
> **Q1.4.** Human expert review.
>
> A total of five experts reviewed and refined the generated entries, with each expert revising between 450-800 entries. Consistency was effectively ensured because the expert editing process primarily involved deleting vague content and reorganizing existing information into the defined Chain-of-Clues format, rather than introducing new information. We also provided strict guidelines on the CoC format. Furthermore, for accuracy, 618 entries selected for evaluation were cross-validated by more than two experts.
>
> **Q1.5.** Dataset statistics.
>
> We appreciate your suggestion about the statistics of our dataset. While **Appendix Table 7** already provides statistics on the number of samples in different splits, we agree that the composition was not detailed. For our ForgeryAnalysis-SFT and Eval datasets, samples are selected proportionally from MIML, CASIA2, DEFACTO, and AutoSplice datasets. The larger ForgeryAnalysis-PT dataset includes 30k, 4k, 13k, and 3k samples, respectively, from these sources. We have now supplemented this by including a statistics table for the source dataset of ForgeryAnalysis in **Appendix A.3**.
>
> | Dataset | Authentic Samples | Tampered Samples | Manipulation Types | Source |
> | --- | --- | --- | --- | --- |
> | MIML [34] | 11,142 (PSBattles) | 123,150 | Manual Editing | PhotoshopBattles |
> | CASIA2 [35] | 7,491 | 5,063 | Splice, Copy-move | Corel |
> | DEFACTO [36] | - | 149,587 | Splice, Copy-move, AIGC (GAN) | MS COCO |
> | AutoSplice [37] | 2,273 | 3,621 | AIGC | Visual News |
>
> **Q2. More details on the method.**
>
> **Q2.1.** Training Prompt $x_p$.
>
> We apologize for initially overlooking the inclusion of the training prompt $x_p$. This prompt is the instruction given to guide the model during training. We have now added a clear explanation in **Appendix Section B**, including an example of the prompt we use:
>
> *Is this image authentic, or has it undergone manipulation? If manipulated, show the precise segmentation mask of the manipulation. Please provide the chain-of-clues supporting your detection decision in the following style: # high-level semantic anomalies (such as content contrary to common sense, inciting and misleading content), # middle-level visual defects (such as traces of tampered region or boundary, lighting inconsistency, perspective relationships, and physical constraints) and # low-level pixel statistics (such as noise, color, textural, sharpness, and AI-generation fingerprint), where the high-level anomalies are significant doubts worth attention, and the middle-level and low-level findings are reliable evidence.*
>
> **Q2.2.** CoC usage for training.
>
> The CoC generated by GPT-4o and refined by human experts is used to train the model auto-regressively. This corresponds to the "text $\hat{T}$ that includes reasoning and evidence" mentioned in **Line 168**. We have stated this in the revised text to avoid any confusion.
>
> **Q2.3.** Special tokens.
>
> Regarding [SEG], < REAL >, and < FAKE > tokens mentioned in **Lines 182-183**, these are new and trainable special tokens added to the LLM's vocabulary. When these tokens are added, trainable embeddings are created for them in the LLM's embedding layer. Specifically, $\tilde{h_{seg}}$ is extracted from the LLM's last-layer embedding corresponding to the [SEG] token. This approach is common in visual language model architectures for grounding tasks, such as LISA [12].
>
> **Q2.4.** Fusion mechanism.
>
> Thank you for your suggestion. We have moved its illustrative figure to the main text and will retain detailed explanations in the Appendix with clear guidance.
>
> **Q3. Evaluation and ablation study.**
>
> **Q3.1.** The qualitative output of the text.
>
> We provided various qualitative outputs of different manipulation types, along with comparisons to existing methods, in **Appendix Figures 11-13**. We have added explicit indications in the relevant sections of the main text to ensure readers are directed to these figures.
>
> **Q3.2.** The performance of fine-tuned LLaVA.
>
> When LLaVA (as our data engine) processes images for analysis, its input includes highlighted tampered regions to facilitate the generation of more accurate manipulation analyses. This task, while related, is distinct from the primary manipulation detection task and relatively simpler. We evaluated the STS performance of our data engine (the fine-tuned LLaVA). The results, evaluated using the same setup as **Table 4**, show STS scores of 0.991, 0.976, and 0.990 across three models, indicating the reliability of our data engine.
>
> **Q3.3.** Classification (real/fake) performance.
>
> We agree that including classification (real/fake) performance is important, as STS primarily assesses analysis text quality. We integrated classification as an evaluation dimension (*Recall*) within our ForgeryAnalysis-Eval benchmark, using output special tokens < REAL > and < FAKE >. The relevant results are presented in **Figure 1(b)** and **Figure 4**. Additionally, our method achieves classification accuracies of 0.989 on Columbia and 0.910 on CASIA1, further demonstrating its robust detection capability.
>
> **Q3.4.** More ablation experiments.
>
> We appreciate your suggestion for further ablation. In our architecture, the decoder requires the LLM's outputted [SEG] token (specifically its $h_{seg}$ embedding) to guide segmentation, preventing us from completely removing the LLM's supervision. However, to achieve a similar validation purpose, we removed all clues, providing only simple text about detection results and [SEG] token. We explored two experimental settings:
>
> **Setting 1 (Only Detection Result):** We removed all clues, providing simple text about detection results (e.g., "This is a < FAKE > image with modifications detected in the [SEG]." or "This is a < REAL > image without modifications detected, and the [SEG] is meaningless.").
>
> **Setting 2 (Unstructured Clues):** We provided summarized text and clues without organizing them into the CoC structure. The order of different clues was randomized.
>
> We used the same evaluation setup as described in **Table 5** in the main text, with *all-mpnet-base-v2* for STS calculation.
>
> | Setting | NIST16 | IMD20 | COCOGlide | ForgeryAnalysis-Eval | STS |
> | --- | --- | --- | --- | --- | --- |
> | ForgerySleuth | 0.518 | 0.710 | 0.562 | 9.45 | 0.961 |
> | Setting 1 | 0.448 | 0.526 | 0.533 | - | - |
> | Setting 2 | 0.505 | 0.681 | 0.564 | 7.94 | 0.785 |
>
> These results demonstrate the effectiveness of the CoC structure in guiding the model towards more precise localization and explanations. Furthermore, if the LLM's output cues are absent, the segmentation performance also declines.
>
> **Q3.5. Failure cases analysis.**
>
> We appreciate your suggestion for an expanded failure case analysis. While the original paper showcased two representative failure cases, we have expanded the failure case analysis in **Appendix Section D.5** as follows, with additional examples for each primary error pattern.
>
> *We analyzed ForgerySleuth's detection failures and identified two representative error patterns:*
>
> *a. Semantic Description Inaccuracies: In these cases, our model can accurately localize manipulated regions, but its textual analysis contains description errors. For instance, in the first example in Appendix Figure 9, the model misidentifies dogs as cats. Crucially, this does not impact the detection logic or localization accuracy. These errors often arise when specific semantic concepts in images are hard to detect clearly. Future work could improve this by enhancing the visual understanding capabilities of MLLM and LLM backbone models.*
>
> *b. Failures with Small Manipulations in Complex Scenes: In complex image content with small manipulated regions, the model's localization precision can drop, sometimes leading to incorrect textual analysis and, consequently, a detection failure. For example, in the second instance in Appendix Figure 9, a spliced cartoon character is present. Due to its small proportion and the image's overall complexity, the model only localized a part of the manipulation (like the guitar in hand) while overlooking the cartoon character. Detecting small region manipulations remains a significant challenge, and MLLMs tend to focus on global semantic information rather than fine-grained local details. Therefore, developing methods to effectively detect anomalies in small regions is a crucial research direction.*

---

### Official Review · Reviewer_w2m9 · 2025-06-16

**Clarity:** 2
**Significance:** 2
**Originality:** 2
**Rating:** 4
**Confidence:** 4

**Summary:**

This paper proposes ForgerySleuth, an image manipulation detection method that leverages the capabilities of VLMs to achieve detection, localization, and explanation of manipulated content. To enable explanatory capabilities for tampered areas, the authors construct a specialized dataset using GPT-4o for both pretraining and evaluation. Experimental results demonstrate that the proposed method achieves effective performance across detection, localization, and explanation tasks.

**Questions:**

1.  The main concerns are shown in the weakness section. Please address the questions about the dataset and novelty.

2. Did you test the contributions of each semantic level? I am curious about the individual contributions of low, middle, and high levels to ForgerySleuth's performance.

3. Did ForgerySleuth undergo testing on cross-domain tasks, like image deepfake detection and localization (face and AIGC)? Experiments in these areas might improve the effectiveness and generalizability of the proposed work.

4. I am concerned about the implementation details. For example, why are λbce and λdice set to 1.0 and 0.2, respectively? Any insights or ablation experiments could support these parameter choices.

5. I advise the authors to include GT masks in the examples (Appendix) to better demonstrate the differences between ForgerySleuth and other methods.

**Ethical Concerns:**

["NO or VERY MINOR ethics concerns only"]

**Final Justification:**

The authors solve most of my questions so I intend to accept.

**Limitations:**

yes

**Quality:**

2

**Strengths And Weaknesses:**

**Strengths**
- The paper proposes an interesting and important topic: providing explanations for manipulation detection. This is particularly relevant as visual models become more and more powerful.
- The experiments demonstrate that the proposed method achieves superior results compared to traditional IFDL methods and Large Language Models (as shown in the Appendix).

**Weaknesses**

- Novelty of Method: The method used in ForgerySleuth lacks complete novelty, as demonstrated by similar approaches in [1,2] and FakeShield, which is discussed in the Appendix. All these methods utilize LLMs to provide explanations for deepfake detection and localization using similar frameworks (e.g., employing SEG tokens to generate masks for manipulation areas, following LISA). The differences between ForgerySleuth and these existing methods are not clearly articulated. A more comprehensive literature review and detailed comparison with existing methods would be helpful to better understand the proposed method's unique advantages.

- Dataset: I'm concern about the progress of building the dataset.
  1. The details regarding expert involvement in the GPT-4o data review process are not provided. This process is quite important since it serves as the foundation for generating the larger ForgeryAnalysis-PT dataset.

  2. In Figure 2(c), the ground truth (GT) masks are not input to the Forgery-Analyst. Instead, the authors choose to use "images with tampered regions highlighted." What does "highlighted" mean in this context, and why are GT masks not used instead? What are the differences between these two approaches? I did not find a detailed explanation of this design choice in the paper.

  3. There are some unclear elements in Figure 2. The figure suggests that Forgery-Analyst is also GPT-4o (though it is not), which creates confusion about the model architecture.
  4. Since the ForgeryAnalysis Dataset utilizes public data, the authors should provide detailed information about the source datasets used, presented in a table or figure. This is necessary for fair comparison with other methods evaluated on public datasets.

[1] Kang, Hengrui et al. “LEGION: Learning to Ground and Explain for Synthetic Image Detection.” ArXiv abs/2503.15264 (2025): n. pag.

[2]  Huang, Zhenglin et al. “SIDA: Social Media Image Deepfake Detection, Localization and Explanation with Large Multimodal Model.” ArXiv abs/2412.04292 (2024): n. pag.

---

> ### Author Rebuttal · Authors · 2025-07-31
>
> We sincerely appreciate your review and constructive suggestions. We apologize for any lack of clarity in the initial submission. We have carefully addressed each of your concerns and questions, and we hope our responses resolve the issues you raised.
>
> **Q1. Novelty of Method.**
>
> We appreciate you pointing out these relevant works. Based on your feedback, we have expanded **Section 2** to better articulate ForgerySleuth's novelties and distinctions.
>
> (1) **Manipulation Type:** While LEGION and SIDA are designed for deepfake detection, focusing on traces left by generative models. Our work and FakeShield address local manipulation, where only a part of the image is edited. These edits can include splicing, manual alterations, or regional generation. Therefore, our work and FakeShield are more relevant in terms of task scope, which is why we focused on comparing with it in the Appendix.
>
> (2) **Model Architecture:** While the overall framework integrating an LLM with a vision segmentation head appears similar (as this design naturally aligns with the task), our core innovation is the motivated and precise design of the LLM addressing high-level semantic anomalies, and the trace encoder capturing low-level statistical features. Furthermore, our fusion mechanism allows these distinct features to complement each other. The effectiveness of these components is further demonstrated by ablation studies in Section 4.x and Q3 below.
>
> (3) **Structured CoC**: Unlike other methods that simply list clues in their datasets, we organized clues into a hierarchical CoC structure during data construction. This not only offers users more coherent analysis but also guides the model to focus on distinct clue levels during training, thereby serving as more effective supervision. The effectiveness of CoC is further demonstrated by new ablation studies in Q3 below.
>
> In summary, our work goes beyond merely integrating an LLM with a segmentation head. We specifically designed our data and model components to leverage distinct forensic clues for image manipulation detection. Furthermore, it is important to note that **our paper was publicly released before LEGION and SIDA, and our model/code before FakeShield.**
>
> **Q2. More details on dataset construction.**
>
> We sincerely appreciate your detailed questions and constructive suggestions regarding our dataset construction. We acknowledge that, due to space constraints, many dataset building details were primarily placed in **Appendix Section A**, leading to less clear representation in the main text. We've carefully considered each of your points and have made the following responses and revisions:
>
> **Q2.1.** Details on expert involvement and review process.
>
> As mentioned in **Lines 141-147**, the experts' review process primarily involved three steps: filtering vague responses, deleting ambiguous clues, and organizing the analysis into the Chain-of-Clues format. We have provided a complete example of this editing process in **Appendix Figure 14**, with more detailed descriptions of the review guidelines available in **Appendix Lines 556-565**. We have now added explicit instructions in the main text to guide readers to this example for better understanding.
>
> **Q2.2.** Explanation of “images with tampered regions highlighted”.
>
> We apologize for the oversight regarding the detailed explanation of this design choice. We have added the following clarification in **Appendix Section A.2**:
>
> *We chose to indicate manipulation by highlighting tampered regions. Here, "highlighting" means outlining the boundaries of the tampered region with a selected color (red, green, or blue, selected based on the least dominant channel in the current image's RGB to ensure visual prominence and distinction from content). The boundary is expanded outwards by 11 pixels to prevent the highlight line from obscuring subtle tampering clues.*
>
> *We did not provide the image and mask as two separate inputs because our experiments showed that MLLMs (like GPT-4o and LLaVA in our method) struggled to accurately correlate corresponding positions between two input images, while annotating directly on a single image is more accurate. **Appendix Figure 19** further illustrates our data engine's input and output examples, including this highlighting method.*
>
> **Q2.3.** Unclear elements in Figure 2.
>
> We apologize for the confusion caused by the ambiguous elements in **Figure 2**. It seems the consistent use of green for both the GPT-4o logo and our data engine might have led to misinterpretations. We've updated **Figure 2(c)** to directly label the model architecture as *LLaVA-13B* rather than *Data Engine*, which should avoid any misunderstanding regarding the model structure.
>
> **Q2.4.** Source datasets used.
>
> We appreciate you raising this point, as clarity on source datasets is crucial for fair comparison. The “existing public datasets” mentioned in **Line 161** refer to the source datasets introduced in **Lines 124-128**. These source datasets are completely independent from our evaluation benchmark datasets (**Lines 246-250**), ensuring fair assessment. We acknowledge that this brief mention was not clear. We have now revised **Line 161** and included a new table to provide detailed statistics on the scale, source, and manipulation types from each public dataset used to construct ForgeryAnalysis and evaluation.
>
> | Dataset | Usage | Authentic Samples | Tampered Samples | Manipulation Types | Source |
> | --- | --- | --- | --- | --- | --- |
> | MIML [34] | Train, Construct ForgerAnalysis | 11,142 (PSBattles) | 123,150 | Manual Editing | PhotoshopBattles |
> | CASIA2 [35] | Train, Construct ForgerAnalysis | 7,491 | 5,063 | Splice, Copy-move | Corel |
> | DEFACTO [36] | Train, Construct ForgerAnalysis | - | 149,587 | Splice, Copy-move, AIGC (GAN) | MS COCO |
> | AutoSplice [37] | Train, Construct ForgerAnalysis | 2,273 | 3,621 | AIGC | Visual News |
> | Columbia [55] | Eval | 183 | 180 | Splice | - |
> | Coverage [56] | Eval | 100 | 100 | Copy-move | - |
> | CASIA1 [35] | Eval | 800 | 920 | Splice, Copy-move | Corel |
> | NIST16 [57] | Eval | - | 564 | Splice, Copy-move | - |
> | IMD20 [58] | Eval | 414 | 2010 | Manual Editing | - |
> | COCOGlide [30] | Eval | 512 | 512 | AIGC (Diffusion) | - |
>
> **Q3. Contributions of each semantic level.**
>
> We appreciate this insightful suggestion, which prompted us to conduct additional ablation experiments to further illustrate the contributions of different semantic levels to ForgerySleuth's performance. Given the extensive data processing and model training required, we have completed the following experiments.
>
> We modified the ForgeryAnalysis-PT and SFT datasets for these specific tests, while the base pre-training process and evaluation datasets remained unchanged. Our experimental settings include:
>
> **Setting 1 (*w/o* High-Level Clues):** We only removed high-level clues from the textual analysis.
>
> **Setting 2 (*w/o* Low-Level Clues):** We only removed low-level clues from the textual analysis.
>
> **Setting 3 (*w/o* Low-Level Clues and Trace Encoder):** We removed low-level clues from the textual analysis and also removed the Trace Encoder.
>
> The evaluation metrics are consistent with **Table 5** in the main text, with the addition of STS calculated using the *all-mpnet-base-v2* model.
>
> | Setting | NIST16 | IMD20 | COCOGlide | ForgeryAnalysis-Eval | STS |
> | --- | --- | --- | --- | --- | --- |
> | ForgerySleuth | 0.518 | 0.710 | 0.562 | 9.45 | 0.961 |
> | Setting 1 (*w/o* High-Level Clues) | 0.493 | 0.551 | 0.548 | 6.31 | 0.764 |
> | Setting 2 (*w/o* Low-Level Clues) | 0.513 | 0.685 | 0.557 | 7.12 | 0.824 |
> | Setting 3 (*w/o* Low-Level Clues and Trace Encoder) | 0.336 | 0.620 | 0.409 | 5.88 | 0.576 |
>
> From these experiments, it can be observed that high-level features contribute relatively more to the manipulation analysis. Removing only low-level features from the textual analysis while retaining the Trace Encoder impacts the comprehensiveness of the manipulation analysis, but has a smaller effect on localization accuracy. In contrast, removing both low-level features from the text and the Trace Encoder significantly degrades localization performance. This further validates the effectiveness of the Trace Encoder in the manipulation detection task.
>
> **Q4. Evaluation on cross-domain tasks.**
>
> We appreciate this important question regarding cross-domain generalization, particularly for deepfake and AIGC content.
>
> While our primary focus is on local image manipulation detection (distinct from full-image deepfake generation as explored by some methods), our evaluation inherently includes relevant cross-domain elements. Our existing evaluation dataset on COCOGlide already incorporates images edited using AIGC methods, and this dataset contains examples of partial face deepfake manipulations (as shown in **Appendix Figure 18**).
>
> **Q5. Implementation details.**
>
> We appreciate your question regarding the implementation details, specifically the weights for $\lambda_{bce}$ and $\lambda_{dice}$. Our initial hyperparameter choices were informed by best practices from relevant works in the field, such as LISA, and LLaVA. We then conducted further experimentation to refine these parameters.
>
> For the two segmentation loss weights ($\lambda_{bce}$ and $\lambda_{dice}$), BCE loss typically serves as the primary loss, while Dice loss is used to enhance performance on imbalanced classes. Therefore, $\lambda_{bce}$ was set to 1.0. We then experimented with various values for $\lambda_{dice}$ (0.1, 0.2, 0.3, 0.5, 1.0) and selected 0.2 based on its optimal performance in the segmentation task.
>
> **Q6. Include GT masks.**
>
> Thank you for this valuable suggestion. We agree that including GT masks will enhance the clarity of our qualitative examples. We have updated figures in the appendix to include GT masks alongside our predictions and those of other methods, providing a clearer visual comparison of localization accuracy.

---

> ### Comment · Reviewer_w2m9 · 2025-08-01
> **Thanks for the response**
>
> Thanks for the detailed response from the authors. I have carefully reviewed both my and the responses from other reviewers, and I'm pleased to see that most of my concerns have been addressed. However, I would like to clarify a few remaining points:
>
> 1. Could you please clarify whether IMD and IFDL refer to the same task, or if there are distinctions between them?
>
> 2. While several works, including ForgerySleuth, employ language models such as GPT to generate explanations, I remain concerned about treating these AI-generated explanations as "ground truth". Have the authors considered alternative approaches for establishing more robust ground truth for this type of task?
>
> Overall, I appreciate the authors' thorough work and thoughtful responses. I will discuss with fellow reviewers and the AC before making my final recommendation.

---

> ### Author Response · Authors · 2025-08-02
>
> Thank you very much for your careful review and positive feedback. We are honored to have this opportunity for a discussion, as your insights are crucial for the further improvement of our paper.
>
> We are glad that our previous response addressed most of your concerns. Regarding the two remaining questions, we would like to provide further clarification.
>
> ---
>
> **Q1. Clarifying IMD and IFDL.**
>
> In the field of image forensics, Image Manipulation Detection (**IMD**) and Image Forgery Detection and Localization (**IFDL**) are often used to describe the **same task**. The difference lies not in the task itself, but may lie in the focus. IMD is the more general term, while IFDL is a more detailed description that emphasizes the dual requirements of both detection and localization.
>
> As we discussed in Q1.(1) of our previous response, for both IMD and IFDL, the key challenge is identifying local manipulations. This task is not limited by the image category or the type of manipulation (e.g., splicing, copy-move, or AIGC editing). The task requires both a classification of whether a manipulation occurred and a precise localization of the tampered region. This corresponds to an image-level classification task and a pixel-level segmentation task, respectively.
>
> Our method, ForgerySleuth, addresses this core task while also introducing a crucial dimension: explanation. Since our approach covers detection, localization, and explanation, we chose the broader IMD term to describe our work. Our model indicates detection results with the < Real > and < Fake > special tokens, and it provides localization results with a segmentation mask. We have thoroughly evaluated our performance across both of these dimensions.
>
> ---
>
> **Q2. Concerns about using AI-Generated explanations as Ground Truth.**
>
> We completely understand your concerns about this matter. It is indeed a key challenge when constructing datasets with large models. We also attempted to create a purely human-annotated dataset, but found that this alternative approach presented significant challenges:
>
> **(1) Difficulty in ensuring expert consistency and completeness.** If experts were to summarize forensic clues from scratch, the analysis would likely vary due to different knowledge focuses. Furthermore, it is particularly challenging for humans to comprehensively identify and summarize certain lower-level clues like lighting inconsistencies, noise statistics, and other subtle artifacts.
>
> **(2) High cost of creating large-scale datasets.** The cost of crafting detailed textual explanations is far higher than annotations for traditional vision tasks (e.g., classification, segmentation). This is not just because of the time required for initial text creation, but also because there are no clear, single annotation rules, demanding experts with extensive forensic experience to identify clues at different levels.
>
> For these reasons, we chose to employ an AI-assisted approach to generate initial explanations. However, unlike methods that rely solely on MLLM, we implemented additional measures to maximize accuracy and minimize hallucinations:
>
> **(1) Expert review and secondary validation mechanism**. We treat the initial data generated by GPT-4o as a "draft," not the final ground truth. Multiple experts review and edit this data, transforming the creative process into one of deleting errors and correcting ambiguities, which not only increases efficiency but also ensures the consistency of the information. For entries used in our evaluation, we specifically required cross-validation by more than one expert to further guarantee their accuracy.
>
> **(2) Specialized data engine model.** We utilized a dedicated data engine model that, during the entire data generation process, is aware of the specific tampered regions as a reference. This allows the model to localize and analyze clues more precisely, enabling us to efficiently expand the dataset while maintaining high quality.
>
> **(3) Constrained and logical CoC structure.** By guiding the MLLM's analysis structure, we ensure its generated text adheres to our proposed Chain-of-Clues (CoC) structure. This not only standardizes the model's output but, more importantly, deconstructs complex forensic analysis into hierarchical clues.
>
> In conclusion, we have fully leveraged existing resources and technology, using an "AI generation and expert review" model to achieve the best possible balance between dataset creation efficiency and quality. We believe that we also provide a crucial supplement to the evaluation of textual explanations, forming a multi-dimensional validation of our model's capabilities, by focusing on quantitative metrics like detection and localization.
>
> ---
>
> Thank you again for the valuable time and effort you have dedicated to this review. It has made our paper more complete and clearer. We sincerely welcome any further questions and suggestions, and genuinely hope you will reconsider your score.

---

> > ### Comment · Reviewer_w2m9 · 2025-08-05
> > **Thanks**
> >
> > Thank you for the additional discussion and clarifications. I’m glad to see you have carefully considered the ground-truth issues and addressed my remaining concerns. Based on the new evidence, I intend to recommend acceptance for now. Please incorporate all experiments, comparisons, and discussions into the new manuscript, as these additions will benefit readers and the community. Good luck for your submission.

---

> > > ### Author Response · Authors · 2025-08-05
> > >
> > > We are truly delighted that our clarifications have addressed your concerns. Your valuable suggestions throughout this process have significantly improved our paper. We will incorporate the discussion into the final manuscript as you recommended.
> > >
> > > Thank you once again for your time and patience during the review process.

---

### Official Review · Reviewer_sU1j · 2025-07-03

**Clarity:** 3
**Significance:** 4
**Originality:** 3
**Rating:** 4
**Confidence:** 3

**Summary:**

The paper presents ForgerySleuth, a framework that uses multimodal large language models (M-LLMs) to detect and explain image manipulations. It combines high-level reasoning from the M-LLM with low-level visual features from a trace encoder to produce both segmentation masks and textual analysis. The authors also built a dataset, ForgeryAnalysis, using a Chain-of-Clues prompting method and expert refinement. Experiments show improved performance over existing methods in both detection accuracy and explanation quality.

**Questions:**

1. The main paper does not clearly specify the dimensions and spatial shapes of key features such as $h_{seg}$, $f_t$, and $f_c$. It would improve clarity if the authors could explicitly provide this information in the main text.

1. While the Chain-of-Clues prompting is central to the method, it is unclear how much it contributes compared to simpler reasoning instructions. Is it possible to provide ablation results or analysis comparing with non-CoC-based annotations?

**Ethical Concerns:**

["NO or VERY MINOR ethics concerns only"]

**Final Justification:**

The rebuttal and follow-up responses addressed my main concerns. The authors clarified key implementation details (e.g., feature dimensions and alignment), provided new ablation results demonstrating the value of Chain-of-Clues prompting, and offered a more comprehensive discussion of limitations and failure cases. These updates improve the clarity and completeness of the paper.

That said, I still find the technical novelty to be somewhat limited, as the approach mainly builds on existing components. Overall, I see this as a solid application-driven contribution with strong experimental support, and I maintain my rating accordingly.

**Limitations:**

Yes

**Paper Formatting Concerns:**

There is a minor inconsistency in the mathematical notation on page 6 (line 204): the text refers to the language model head output as $\hat{y}_{txt}$, while the formula uses $\hat{T}$. These likely represent the same output, but the discrepancy could confuse readers. Clarifying or unifying these symbols would improve rigor.

**Quality:**

3

**Strengths And Weaknesses:**

## Strengths
- The paper explores a novel application of M-LLMs in image manipulation detection, a relatively underexplored area.
- The proposed system effectively combines high-level semantic reasoning with low-level trace features, leading to strong performance in both localization and explanation.
- The Chain-of-Clues structure improves interpretability and aligns well with the needs of forensic analysis.
- The ForgeryAnalysis dataset is carefully constructed, with expert refinement and scalable pretraining data, supporting both accuracy and explainability.
- Experimental results are comprehensive, covering multiple benchmarks and including both quantitative and qualitative evaluation.

## Weaknesses
- The paper lacks detail on key implementation aspects, such as the dimensions and alignment of feature representations across modules.
- Most components are built on existing methods (e.g., trace encoder, LoRA, attention fusion), and the novelty mainly lies in their combination rather than in any single technical innovation.
- The discussion on limitations is minimal in the main paper, and some design choices (e.g., freezing the vision encoder) are not well justified.

---

> ### Author Rebuttal · Authors · 2025-07-31
>
> We sincerely appreciate your thorough review and positive assessment of our work. We have carefully considered all your questions and suggestions, and our revisions are detailed below.
>
> **Q1. Dimensions and alignment of features**
>
> We appreciate your suggestion for clearer specifications regarding feature dimensions and their alignment. We have revised **Section 4**, specifically **Lines 181-205,** to provide this information.
>
> **Semantic Clue Embedding $h_{seg}$.** Given an input image $x_{img} \in \mathbb{R}^{B \times H \times W \times 3}$ (where $B$ is batch size, and $H$ and $W$ are the height and width of the image respectively) and prompt $x_p$, we first feed them into the M-LLM $F_m$. Fm outputs a hidden embedding $\tilde{h} \in \mathbb{R}^{B×L×\tilde{D_{m}}}$ from its last layer (where $L$ is sequence length, and $\tilde{D_{m}}$ is the hidden layers' dimension of the LLM). We then extract the embedding $\tilde{h_{seg}} \in \mathbb{R}^{B×\tilde{D_{m}}}$ corresponding to the [SEG] token from $\tilde{h}$ and apply an MLP projection layer $\gamma$ to obtain $h_{seg} \in \mathbb{R}^{B×D_{m}}$. This $h_{seg}$ serves as the semantic clue embedding that guides mask generation.
>
> **Trace Feature $f_t$.** Simultaneously, the input image $x_{img}$ is processed by the trace encoder $F_t$. This encoder extracts low-level forensic traces, yielding manipulation trace features $f_t \in \mathbb{R}^{B \times H_t \times W_t \times D_t}$, where $H_t = H / \text{patch.size}$ and $W_t = W / \text{patch.size}$.
>
> **Content Feature $f_c$.** The vision backbone $F_v$ processes $x_{img}$ to produce a dense vision content feature $f_c \in \mathbb{R}^{B \times H_c \times W_c \times D_c}$, where $H_c = H / \text{patch.size}$ and $W_c = W / \text{patch.size}$.
>
> For effective multimodal fusion, the dimensions $D_m$, $D_t$, and $D_c$ are projected to a common embedding dimension $D$ before they are fed into the fusion mechanism.
>
> **Q2. Technical innovation.**
>
> We acknowledge that ForgerySleuth integrates existing methods such as LLMs, LoRA, and attention fusion. While such combinations are common in application-oriented solutions, our core innovation lies in the motivation and precise design of how the LLM addresses high-level semantic anomalies while the trace encoder captures low-level statistical features. Experiments further demonstrate the effectiveness of these components, including our method and dataset.
>
> **Q3.  Ablation study of CoC prompting.**
>
> We appreciate your suggestion for an ablation study on CoC prompting, which effectively clarifies its contribution. To demonstrate how much CoC contributes compared to simpler reasoning instructions, we have conducted new ablation experiments using non-CoC-based annotations. We explored two experimental settings:
>
> **Setting 1 (Only Detection Result):** We removed all clues, providing only simple text about detection results (e.g., "This is a <FAKE> image with modifications detected in the [SEG]." or "This is a <REAL> image without modifications detected, and the [SEG] is meaningless.").
>
> **Setting 2 (Unstructured Clues):** We provided summarized text and clues without organizing them into the CoC structure. The order of different clues was randomized.
>
> We employed the same evaluation setup as described in **Table 5** in the main text, with *all-mpnet-base-v2* for STS calculation. The results are presented in the table below:
>
> | Setting | NIST16 | IMD20 | COCOGlide | ForgeryAnalysis-Eval | STS |
> | --- | --- | --- | --- | --- | --- |
> | ForgerySleuth (with CoC) | 0.518 | 0.710 | 0.562 | 9.45 | 0.961 |
> | Setting 1 (Only Detection Result) | 0.448 | 0.526 | 0.533 | - | - |
> | Setting 2 (Unstructured Clues) | 0.505 | 0.681 | 0.564 | 7.94 | 0.785 |
>
> These results demonstrate the effectiveness of the CoC structure in guiding the model towards more precise localization and coherent, structured explanations.
>
> **Q4. Justification for freezing the vision encoder.**
>
> We appreciate your question regarding the justification for freezing the vision encoder. The primary reason for freezing the vision backbone $F_v$ is to retain its robust capacity for modeling image content features, which are crucial for accurate segmentation. This design choice is detailed in **Lines 227-229**.
>
> Furthermore, we have validated this decision through ablation experiments, as presented in **Lines 328-331** and **Table 6**. Our comparison of various training strategies for the vision backbone $F_v$, including trainable and LoRA fine-tuning, consistently demonstrated that keeping the encoder completely frozen achieves the best performance. This can be attributed to the observation that trainable and LoRA fine-tuned strategies slightly diminish the generalization ability of the original SAM backbone.
>
> **Q5. Limitations.**
>
> We appreciate you highlighting the limited scope of our initial limitations discussion. We have accordingly revised and expanded **Appendix Section D.6 (Limitation and Future Work)** in the paper, integrating the failure case analysis (**Appendix Section D.5**) for a more comprehensive discussion. Our updated limitations are as follows:
>
> ***1. Challenges in Detection Performance.** We analyzed ForgerySleuth's detection failures and identified two representative error patterns:*
>
> *a. Semantic Description Inaccuracies: In these cases, our model can accurately localize manipulated regions, but its textual analysis contains description errors. For instance, in the first example in Appendix Figure 9, the model misidentifies dogs as cats. Importantly, this does not impact the detection logic or localization accuracy. These errors often arise when specific semantic concepts in images are challenging to detect clearly. Future work could improve this by enhancing the visual understanding capabilities of MLLM and LLM backbone models.*
>
> *b. Failures with Small Manipulations in Complex Scenes: In complex image content with small manipulated regions, the model's localization precision can degrade, sometimes leading to incorrect textual analysis and, consequently, a detection failure. For example, in the second instance in Appendix Figure 9, where a spliced cartoon character is present, the model only localized a partial manipulation (e.g., the guitar in hand) while overlooking the character itself, primarily due to the small proportion of the manipulation and the image's overall complexity. Detecting small region manipulations remains a significant challenge, and MLLMs tend to focus on global semantic information rather than fine-grained local details. Therefore, developing methods to effectively detect anomalies in small regions is a crucial research direction.*
>
> ***2. Constraints in Evaluation Methods.** Our paper evaluates analysis accuracy using GPT-4o and STS methods on the ForgeryAnalysis-eval dataset. However, our evaluation methods are subject to specific constraints: our assessment relies on GPT-4o, a large external model that may exhibit inherent biases or hallucinations, and STS only reflects semantic similarity between two sentences. Thus, exploring more comprehensive and objective evaluation methods, including the quantitative assessment of model hallucinations, is an important direction for future work.*
>
> ***3. Model Scale and Inference Costs.** While leveraging LLMs to complement traditional image manipulation detection methods, which often primarily focus on low-level traces, this approach increases the model's scale and computational cost of generating detailed explanations. In the future, we plan to investigate model light-weighting solutions. This includes developing a version where generating such detailed output is optional, which would significantly reduce inference time. Alternatively, we could explore distilling LLM capabilities into smaller models, retaining only the essential knowledge for manipulation detection.*

---

> > ### Comment · Reviewer_sU1j · 2025-08-02
> > **Acknowledgement of rebuttal**
> >
> > Thank you for the detailed rebuttal and additional experiments.
> >
> > Your clarifications on feature dimensions and the Chain-of-Clues prompting ablation addressed my questions thoroughly. I also appreciate the expanded discussion of limitations. I also realize that I had previously overlooked the justification you already provided regarding freezing the vision encoder - apologies for the oversight.
> >
> > I have no further questions and will take these updates into account when finalizing my review.

---

> ### Author Response · Authors · 2025-08-02
>
> Thank you so much for your time in reviewing our rebuttal and the additional experiments.
>
> We are very pleased that our clarifications and revisions have successfully addressed your questions. We also truly appreciate your positive feedback and understanding of our work. The confusion you highlighted and your valuable suggestions were crucial in helping us improve our paper.
>
> We are always open to further discussion if you have any more questions. We look forward to your final recommendation.

---

### Official Review · Reviewer_i8FQ · 2025-07-05

**Clarity:** 2
**Significance:** 3
**Originality:** 3
**Rating:** 4
**Confidence:** 3

**Summary:**

In this work, the authors explore the image manipulation detection tasks using multimodal large language models. They propose ForgerySleuth, which integrates MLLMs with a trace encoder to use the world knowledge to capture high-level semantic anomalies as well as low-level traces. They also introduced a supervised fine-tuning dataset, ForgeryAnalysis, designed to pre-training and fine-tuning for IMD task, as well as a data engine to facilitate the creation of a larger-scale ForgeryAnalysis-PT dataset for pre-training. The author conduct experimentations on 6 different datasets, including Columbia, Coverage, CASIA1, NIST16, IMD20, and COCOGlide, and demonstrate superior performance. Code is also publicly available.

**Questions:**

1. Need more details on dataset curation and construction. For example, how did you edit the image? what are the process that leads to modification of a specific part of the image? Do you also have positive examples in your dataset?
Also please add more explanation to the caption of figure 2.
2. While the method is well-motivated for real-world usage, the paper lacks practical demonstrations. Could the authors provide a proof-of-concept demo or case study (e.g., detecting social media tampering or forensic usage) to better contextualize the real-world relevance?
3. The supervised fine-tuning and evaluation datasets seem limited and potentially overlapping. Can the authors clarify how they ensured no data leakage between fine-tuning and evaluation sets? Have they tested generalization to out-of-domain manipulations or unseen manipulation types?
4. Can you provide more details on the data engine you use?

**Ethical Concerns:**

["NO or VERY MINOR ethics concerns only"]

**Final Justification:**

Based on the rebuttal, I'm satisfied with the response. I decide to keep my rating of "Borderline accept".

**Limitations:**

Discussion very limited. Should expand it other than stating "the model is too big" and "create a lighter version and faster inference". There is no insight. Please discuss your limitations. What are some problems or challenges when face during the data generation? Any problem with the quality of the data? What are some patterns of failure you found in your proposed method?

**Quality:**

3

**Strengths And Weaknesses:**

Strength:
1. Paper is well-written and easy to understand in most part. The topic and task of the paper is well motivated and important.
2. The author curate dataset and data curation pipeline, which is useful for future research and manipulation detection applications.
3. Thorough experimentation and ablations. Thorough analysis on different types of manipulations.


Weakness:
1. Need more details on dataset curation and construction. Currently, this section is really unclear. Including the caption of figure 2.
2. It would be nice to discuss and demonstrate some practical usage of your work, or demo.
3. The SFT and eval dataset split is very limited, which is a concern for generalization.
4. poorly-written limitation section.

---

> ### Author Rebuttal · Authors · 2025-07-31
>
> We sincerely appreciate your feedback and positive comments on our work. We have addressed each weakness and question raised, and the paper has been updated and expanded accordingly. Below is a detailed response to your points.
>
> **Q1. More details on dataset construction.**
>
> Thank you for requesting more details on our dataset construction. To clarify, our pipeline does not include the image editing process itself. Instead, we use several public image manipulation detection datasets as our data sources. These datasets already contain real images, manipulated images, and their corresponding manipulation region annotations. A detailed introduction to these sources is provided in **Lines 124-128** of our paper.
>
> We acknowledge that the data production process may require further clarification. While **Appendix Figure 14** already illustrates a complete data production overview, we have added more explicit indicators in the main text to guide readers to this figure and its detailed explanation in the Appendix. Furthermore, as suggested, we have revised the caption for **Figure 2** to enhance clarity. The updated caption reads:
>
> *Figure 2: ForgeryAnalysis Dataset Construction Pipeline. Our pipeline begins with (a) GPT-4o generating initial analyses for manipulated images with annotated regions, followed by human expert review. The refined analyses are organized into (b) the CoC format. This human-curated ForgeryAnalysis (2k) dataset is used to train a data engine. Finally, (c) this data engine generates ForgeryAnalysis-PT, a larger-scale dataset for model pre-training.*
>
> **Q2. Real-world application.**
>
> We agree on the importance of real-world application. We've enhanced the experimental analysis section (**Section 5.2**) to emphasize the following points, which illustrate ForgerySleuth's potential in social media and forensic analysis:
>
> First, a portion of ForgeryAnalysis's source data (e.g., the MIML [34] dataset collected from PhotoshopBattles social platform) comprises images edited by real users for various purposes, commonly found on social media. This represents a typical real-world scenario, indicating our ForgeryAnalysis-eval already encompasses this type of data.
>
> Second, we have further augmented our evaluation with experiments on the GRE [a] dataset. This dataset contains real-world black-box edited data sourced from X (formerly Twitter) and Weibo. On the manipulation region detection task, our method achieved an F1 score of 70.3%. These results further demonstrate ForgerySleuth's generalization in complex real-world scenarios.
>
> *[a] Sun, Z., Fang, H., Cao, J., Zhao, X., & Wang, D. (2024, October). Rethinking image editing detection in the era of generative AI revolution. In Proceedings of the 32nd ACM International Conference on Multimedia (pp. 3538-3547).*
>
> **Q3. The eval split in ForgeryAnalysis.**
>
> We understand your concern regarding the potentially limited size of our ForgeryAnalysis supervised fine-tuning and evaluation datasets. This limitation results from the cost of ensuring high-quality human expert review and cross-validation for accuracy. However, our evaluation extends beyond this single dataset, also including tests on the manipulation region localization task, ensuring comprehensive validation.
>
> We ensured no data leakage between our fine-tuning and evaluation sets by implementing a strict split based on the original image source. Specifically, images derived from the same base image (e.g., different manipulations of the same original photograph) are allocated exclusively to either the training/fine-tuning set or the evaluation set, but never both.
>
> Regarding generalization to out-of-domain or unseen manipulation types, our evaluation includes performance on six distinct benchmark datasets (**Lines 246-250**). These datasets are entirely separate from our training data source (**Lines 124-128**) and feature different manipulation types. This cross-dataset evaluation approach, employing distinct datasets with various manipulation types, is a widely accepted practice in the image manipulation detection field to demonstrate generalization capabilities.
>
> To provide further clarity, we have now included a new table to provide detailed statistics on the scale, source, and manipulation types from each public dataset utilized in the construction of ForgeryAnalysis and for evaluation.
>
> | Dataset | Usage | Authentic Samples | Tampered Samples | Manipulation Types | Source |
> | --- | --- | --- | --- | --- | --- |
> | MIML [34] | Train, Construct ForgerAnalysis | 11,142 (PSBattles) | 123,150 | Manual Editing | PhotoshopBattles |
> | CASIA2 [35] | Train, Construct ForgerAnalysis | 7,491 | 5,063 | Splice, Copy-move | Corel |
> | DEFACTO [36] | Train, Construct ForgerAnalysis | - | 149,587 | Splice, Copy-move, AIGC (GAN) | MS COCO |
> | AutoSplice [37] | Train, Construct ForgerAnalysis | 2,273 | 3,621 | AIGC | Visual News |
> | Columbia [55] | Eval | 183 | 180 | Splice | - |
> | Coverage [56] | Eval | 100 | 100 | Copy-move | - |
> | CASIA1 [35] | Eval | 800 | 920 | Splice, Copy-move | Corel |
> | NIST16 [57] | Eval | - | 564 | Splice, Copy-move | - |
> | IMD20 [58] | Eval | 414 | 2010 | Manual Editing | - |
> | COCOGlide [30] | Eval | 512 | 512 | AIGC (Diffusion) | - |
>
> **Q4. More details on the data engine.**
>
> We regret any lack of clarity regarding our data engine. As detailed in **Appendix Section A.2**, we provided a more comprehensive description of our data engine, including its architecture and prompt design. Additionally, **Appendix Figure 19** already offers specific examples that illustrate the data engine's outputs for better understanding.
>
> Based on your valuable feedback, we recognize that these details might not have been sufficiently emphasized in the main text. Therefore, we have added explicit guidance and incorporated the following content in **Section 3.2** of the main paper:
>
> *The data engine receives input that includes explicit information about the tampered region (highlighted to indicate tampering), aiming to generate more precise and comprehensive clue analyses. It outputs manipulation analyses organized in the CoC format, as illustrated in Figure 2(c). For more detailed information on the data engine, including the specific prompts used, please refer to Appendix Section A.2, and Appendix Figure 19 for output examples.*
>
> **Q5. Limitations.**
>
> We are highly appreciative of your feedback on our limitations section. Your suggestions prompted us to reconsider our work's current limitations and future improvement directions. We have revised and expanded **Appendix Section D.6 (Limitation and Future Work)** in the paper, integrating the failure case analysis (**Appendix Section D.5**) for a more comprehensive discussion. Our updated limitations section is as follows:
>
> ***1. Challenges in Detection Performance.** We analyzed ForgerySleuth's detection failures and identified two representative error patterns:*
>
> *a. Semantic Description Inaccuracies: In these cases, our model can accurately localize manipulated regions, but its textual analysis contains description errors. For instance, in the first example in Appendix Figure 9, the model misidentifies dogs as cats. Importantly, this does not impact the detection logic or localization accuracy. These errors often arise when specific semantic concepts in images are challenging to detect clearly. Future work could improve this by enhancing the visual understanding capabilities of MLLM and LLM backbone models.*
>
> *b. Failures with Small Manipulations in Complex Scenes: In complex image content with small manipulated regions, the model's localization precision can degrade, sometimes leading to incorrect textual analysis and, consequently, a detection failure. For example, in the second instance in Appendix Figure 9, where a spliced cartoon character is present, the model only localized a partial manipulation (e.g., the guitar in hand) while overlooking the character itself, primarily due to the small proportion of the manipulation and the image's overall complexity. Detecting small region manipulations remains a significant challenge, and MLLMs tend to focus on global semantic information rather than fine-grained local details. Therefore, developing methods to effectively detect anomalies in small regions is a crucial research direction.*
>
> ***2. Constraints in Evaluation Methods.** Our paper evaluates analysis accuracy using GPT-4o and STS methods on the ForgeryAnalysis-eval dataset. However, our evaluation methods are subject to specific constraints: our assessment relies on GPT-4o, a large external model that may exhibit inherent biases or hallucinations, and STS only reflects semantic similarity between two sentences. Thus, exploring more comprehensive and objective evaluation methods, including the quantitative assessment of model hallucinations, is an important direction for future work.*
>
> ***3. Model Scale and Inference Costs.** While leveraging LLMs to complement traditional image manipulation detection methods, which often primarily focus on low-level traces, this approach increases the model's scale and computational cost of generating detailed explanations. In the future, we plan to investigate model light-weighting solutions. This includes developing a version where generating such detailed output is optional, which would significantly reduce inference time. Alternatively, we could explore distilling LLM capabilities into smaller models, retaining only the essential knowledge for manipulation detection.*

---

> > ### Comment · Reviewer_i8FQ · 2025-08-05
> >
> > Thanks for the detailed rebuttal. It addressed my concerns and I will take this rebuttal into consideration for my final rating, after seeing all of the comments and discussions.

---

> > > ### Author Response · Authors · 2025-08-05
> > >
> > > Thank you for your follow-up and positive feedback. We are very pleased that our rebuttal addressed your concerns. We are honored to continue the discussion if you have any further questions.
> > >
> > > We want to express our sincere gratitude once again for your time and effort in the review process.

---

### Note · Authors · 2025-08-11

Dear Reviewers and Area Chairs,

We wish to extend our sincere gratitude for your valuable time and insightful suggestions throughout the review and discussion period. We are pleased that our response and the subsequent discussion have effectively clarified and resolved the initial concerns. We are very encouraged by the positive feedback on significance, motivation, novelty, and effectiveness.

The suggestions and discussion have been instrumental in strengthening our paper, and we would like to highlight the key revisions that were discussed:

- Key details initially placed in the appendix have been integrated into the main text to improve clarity. We also provided more explicit links and explanations for the richer information in the appendix, ensuring all content is easy to access.
- The additional ablation studies mentioned during the rebuttal, particularly those on the Chain-of-Clues (CoC), have been integrated into the manuscript. These results provide strong support for our motivation and were recognized as a valuable addition.
- As suggested by the reviewers, we have supplemented the manuscript with the dataset statistics. This addition provides greater transparency and clarity regarding our data.
- We have expanded the discussion on limitations and failure cases. The revised analysis, as provided in our rebuttal, offers a more comprehensive perspective, contributing to a more balanced view of our work.

We also confirm that all other revisions suggested during the discussion have been carefully addressed in the final manuscript.

We thank you once again for your effort and constructive feedback, which have significantly contributed to the improvement of our work.

Best regards,

Authors of Paper 16239

---

### Decision · Program_Chairs · 2025-09-17

**Decision:**

Accept (poster)

**Comment:**

This paper proposes ForgerySleuth, a framework that leverages multimodal large language models to detect, localize, and explain image manipulations, along with a new dataset (ForgeryAnalysis). Reviewers appreciated the paper's clear motivation, thorough experimentation, and the structured approach to interpretability. Reviewers i8FQ and sU1j in particular were excited about the proposed dataset and the effectiveness of the ForgerySleuth method. The chain-of-clues structure was noted as compelling for improving interpretability.

Initial concerns, such as unclear dataset construction, lack of practical examples, and limited discussion of limitations, were largely addressed in the rebuttal through clarifications and additional analysis. While some questions remain about reliance on AI-generated annotations and the degree of novelty, the work is technically solid and contributes meaningfully to an underexplored problem. The paper ended the discussion with four borderline accept ratings, and the AC recommends acceptance.